# Dissection of genetic variation and evidence for pleiotropy in male pattern baldness

Chloe X. Yap [1], Julia Sidorenko[1,2], Yang Wu [1], Kathryn E. Kemper[1], Jian Yang [1,3], Naomi R. Wray [1,3], Matthew R. Robinson[1,4,5] & Peter M. Visscher [1,3]

Male pattern baldness (MPB) is a sex-limited, age-related, complex trait. We study MPB genetics in 205,327 European males from the UK Biobank. Here we show that MPB is strongly heritable and polygenic, with pedigree-heritability of 0.62 (SE = 0.03) estimated from close relatives, and SNP-heritability of 0.39 (SE = 0.01) from conventionally-unrelated males. We detect 624 near-independent genome-wide loci, contributing SNP-heritability of 0.25 (SE = 0.01), of which 26 X-chromosome loci explain 11.6%. Autosomal genetic variance is enriched for common variants and regions of lower linkage disequilibrium. We identify plausible genetic correlations between MPB and multiple sex-limited markers of earlier puberty, increased bone mineral density ($r_g$ = 0.15) and pancreatic β-cell function ($r_g$ = 0.12). Correlations with reproductive traits imply an effect on fitness, consistent with an estimated linear selection gradient of -0.018 per MPB standard deviation. Overall, we provide genetic insights into MPB: a phenotype of interest in its own right, with value as a model sex-limited, complex trait.

[1] Institute for Molecular Bioscience, University of Queensland, Queensland, QLD 4072, Australia. [2] Estonian Genome Center, Institute of Genomics, University of Tartu, 51010 Tartu, Estonia. [3] Queensland Brain Institute, University of Queensland, Queensland, QLD 4072, Australia. [4] Department of Computational Biology, University of Lausanne, 1015 Lausanne, Switzerland. [5] Swiss Institute of Bioinformatics, CH-1015 Lausanne, Switzerland. Correspondence and requests for materials should be addressed to P.M.V. (email: peter.visscher@uq.edu.au)

Male pattern baldness (MPB), or androgenetic alopecia, is an age-related condition characterised by hair thinning, miniaturisation and loss[1] that affects around 80% of European men[2]. The balding process is highly patterned, suggesting that MPB progression is induced by some ageing-associated program: from initial frontotemporal hairline recession to a more severe occipital horseshoe stage[1]. MPB has epidemiological links with conditions including prostate cancer[3–5], cardiometabolic disease[6] and Parkinson's Disease[3]. However, despite the prevalence of MPB, the underlying biology is not well understood.

Current evidence suggests that the genetic contribution to MPB is strong and polygenic. Early twin studies estimated the narrow-sense heritability ($h^2$) on a scale of liability to be 0.81[7] (95%CI: 0.77–0.85). Estimates of variance explained by genome-wide autosomal SNPs ($h^2_{SNP}$) range from 0.34 to 0.67[4,8]. (See Box 1 for an overview of heritability definitions.) The Xq12 locus within the androgen receptor gene (*AR*) was the first locus to be linked to MPB[9], and was replicated across different populations[10], in linkage analysis[11] and in genome-wide association studies (GWAS)[4,8,12–14]. The androgen receptor binds testosterone, and increased sensitivity is associated with more severe MPB[15]. Three large-scale MPB GWAS studies were published in 2017, significantly increasing the number of candidate loci. Heilmann-Heimbach et al.[4] performed a GWAS case-control meta-analysis on over 22,000 males, finding 63 independent MPB-associated loci, with six on the X-chromosome. Three X-chromosome loci had plausible candidate genes, including *AR* (on Xq12), *TSR2* and *KLF8* (both on Xp11.21). Hagenaars et al.[8] used an even larger dataset from the UK Biobank (UKB) pilot phase (UKBv1) of over 52,000 males; they treated MPB as a ordered categorical trait with four levels, and identified 287 independent MPB-associated variants: 247 autosomal, and 40 on the X-chromosome[8]. In a case-control study using the UKBv1 dataset for discovery ($n = 43,590$), Pirastu et al. identified and replicated 107 independent SNPs representing 71 MPB-associated loci (two on the X-chromosome); 30 loci were reported to be novel[16]. All three studies use different definitions for distinct loci, so the reported numbers are difficult to compare.

It is not uncommon for published GWAS to neglect analyses of the X-chromosome, as differential dosage between the sexes presents unique challenges[17]. As MPB is a sex-limited trait, it is notable that the *AR* X-chromosome locus has a particularly large effect size[13], and other MPB-associated loci also converge on androgenetic pathways[15,16,18] (in addition to Wnt signalling and apoptosis[16]). Since the X-chromosome contains 5% of all human genes[17], and many complex diseases (including autoimmune, cardiovascular and psychiatric disorders) are sexually dimorphic with respect to their lifetime risk, age of onset and symptoms[19], it is important to investigate how the X-chromosome contributes to complex traits. Thus, MPB is an interesting trait to study due to the known X-chromosome effect, its sex-limited expression, its complex, polygenic basis, and its manifestation late in life.

Here, we investigate the genetic basis of MPB, drawing upon data from 205,327 adult males of European ancestry from the UKB. We quantify and dissect genetic variation on the autosomes and the X-chromosome, finding $h^2$ as high as 0.62. Our GWAS shows that MPB is highly polygenic, and we also find associations with other sex-limited traits (including female-limited traits) and bone mineral density that suggest pleiotropy.

## Results

**MPB sample characteristics in the UK Biobank.** In the UKB, the self-reported MPB trait was scored along a scale of 1 to 4, representing increasing severity (Fig. 1a, Supplementary Fig. 1). Subjects were aged between 40 and 73, with the distribution skewed towards older ages (Fig. 1b). As expected, individuals with lower MPB scores tended to be younger than those with higher scores (Fig. 1b). The proportions of individuals in each MPB category were 32%, 23%, 27% and 18% for MPB scores 1–4, respectively (Fig. 1c). The regression coefficient of raw MPB scores on age was 0.02 ($P < 2e-16$); that is on average, each year confers a 0.02 increase in MPB score. Including age-squared in the regression did not explain significantly more variance. The estimate of repeatability from longitudinal self-report data from 9603 men was 0.88 (95%CI: 0.87–0.88) (Fig. 2).

**Modelling of the MPB trait.** For all analyses, we took as our trait residuals from regressing the MPB scores on age, assessment centre, ethnicity and 40 principal components calculated using the UKB European sample. The residuals are hereafter referred to as 'adjusted MPB scores' (Fig. 1d), with standard deviation of 1.1.

**Heritability estimation.** First, we estimated pedigree-based $h^2$ ($h^2_{ped}$) using first-degree relatives determined by UKB kinship coefficients, including 4428 full-brothers and 789 father–son pairs. Our estimate was 0.589 (SE = 0.026) for all first-degree relatives, and 0.619 (SE = 0.028) from only brother–brother pairs. In contrast, the estimate from 789 father–son pairs (0.411, SE = 0.071) (Fig. 2a, Supplementary Data 1) was significantly

---

### Box 1: Definitions of heritability

Heritability is a measure of the proportion of phenotypic variance attributed to genetic factors; here, we consider only additive genetic factors (narrow-sense heritability, $h^2$).

$h^2$ is estimated from family data ($h^2_{ped}$) using information from close relatives (e.g. third-degree and closer). However, these estimates may be inflated as relatives may share environmental exposures and non-additive genetic effects. Genetic variance can be measured directly from DNA markers such as SNPs: so-called SNP-based heritability ($h^2_{SNP}$). The expected magnitude of $h^2_{SNP}$ depends on the SNP set used in its estimation. $h^2_{SNP}$ is estimated based on the small genetic relationships between conventionally unrelated individuals (the estimate of $h^2_{SNP}$ is therefore unlikely to be biased by shared environmental exposures or non-additive genetic effects). In this method, relatedness coefficients are calculated based on similarities of alleles between pairs of individuals. Here, we used the GREML method implemented in the GCTA software[20, 21]. Large sample size is required to obtain an estimate of $h^2_{SNP}$ with a relatively small standard error because of the small variation in the genetic relatedness between unrelated individuals. The estimate of $h^2_{SNP}$ tends to be lower than $h^2_{ped}$ because the former is less likely to include variation due to shared environmental effects and rare variants. The estimates of $h^2_{SNP}$ usually rely on similarity of common SNPs, which underestimates $h^2$ if the genetic architecture is dominated by rare variants. The estimates of $h^2_{SNP}$ may be divided into further components[70]. Here, we separated the autosomal and X-chromosome genetic components. We also further compartmentalised the autosomal contribution into (a) low versus high LD and (b) rare versus common SNPs in our GCTA GREML-LDMS analysis[23].

It is also possible to simultaneously estimate $h^2_{SNP}$ and $h^2_{ped}$ using data from both close and distant relatives, in the so-called 'big K small K' analysis[22].

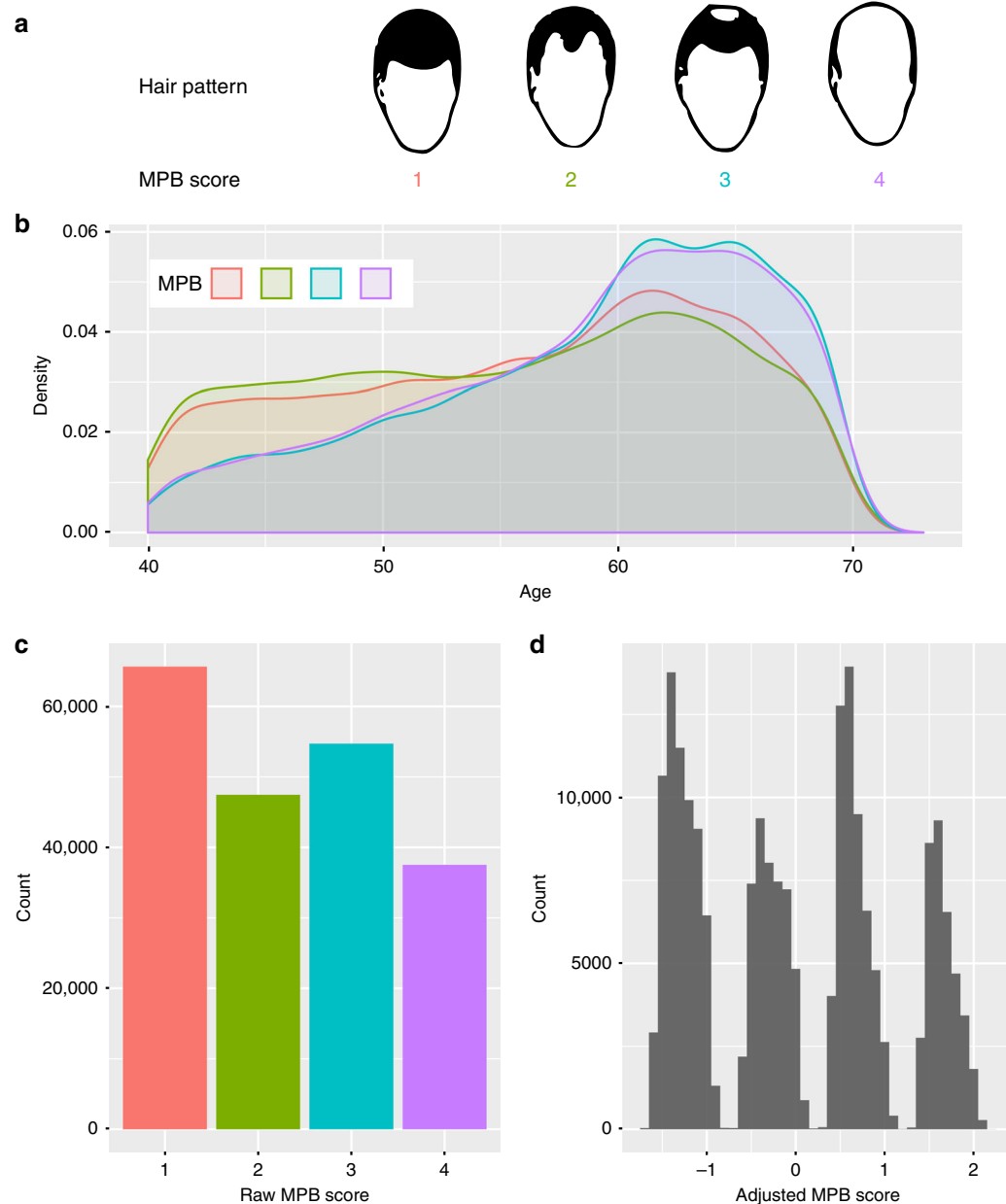

**Fig. 1** Summary of the European, genotyped, genetically-male study population ($n = 205,327$). **a** Diagram of hair patterns and corresponding MPB scores, adapted from the UKB baldness survey, accessible at https://biobank.ctsu.ox.ac.uk/crystal/refer.cgi?id=100423. **b** Density plots showing the relative age distributions within each MPB score group. **c** Distribution of raw MPB scores and **d** adjusted MPB scores (age, assessment centre, ethnicity and 40 principal components calculated using the UKB European population)

lower ($P = 0.006$, two-sided test for the difference between these estimates).

Next, we estimated SNP-based $h^2$ ($h^2_{SNP}$) using pairs of unrelated individuals and GCTA software[20,21]. Due to computational constraints, we defined a sample of 87,957 individuals, which included the maximum number of related pairs (genomic relationship matrix (GRM) coefficient of relationship, rel > 0.05). From this sample, there was a maximum subset of 75,654 unrelated individuals (rel < 0.05) (Fig. 2a, Supplementary Fig. 2). We jointly fitted an autosomal and X-chromosome GRM on these individuals, and estimated total $h^2_{SNP}$ of 0.393 (SE = 0.006), partitioned as 0.359 (SE = 0.006) from the autosomes and 0.034 (SE = 0.002) from the X-chromosome (Fig. 2a, Supplementary Data 1). We confirmed that these partitions made independent

contributions by calculating variances from these GRMs separately (Supplementary Data 1).

We performed a big K small K analysis[22] ($n = 87,957$) based on autosomal data to estimate $h^2_{ped}$ and $h^2_{SNP}$ simultaneously. This gave $h^2_{ped}$ of 0.610 (SE = 0.030); of this, 0.349 (SE = 0.005) was explained by common autosomal SNPs (MAF > 0.01) among unrelated individuals (i.e. $h^2_{SNP}$) and 0.261 (SE = 0.030) was explained by close relatives (big K GRM threshold replacing rel < 0.05 with 0) (Supplementary Data 1). Assuming that this estimate reflects only genetic effects, this indicates that common autosomal SNPs capture 57% of the additive genetic variance. Subsequently, we added a third GRM built using common X-chromosome SNPs (MAF > 0.01). Similarly, total $h^2_{ped}$ was 0.604 (SE = 0.029), partitioned as 0.359 from common autosomal SNPs (i.e. $h^2_{SNP}$,

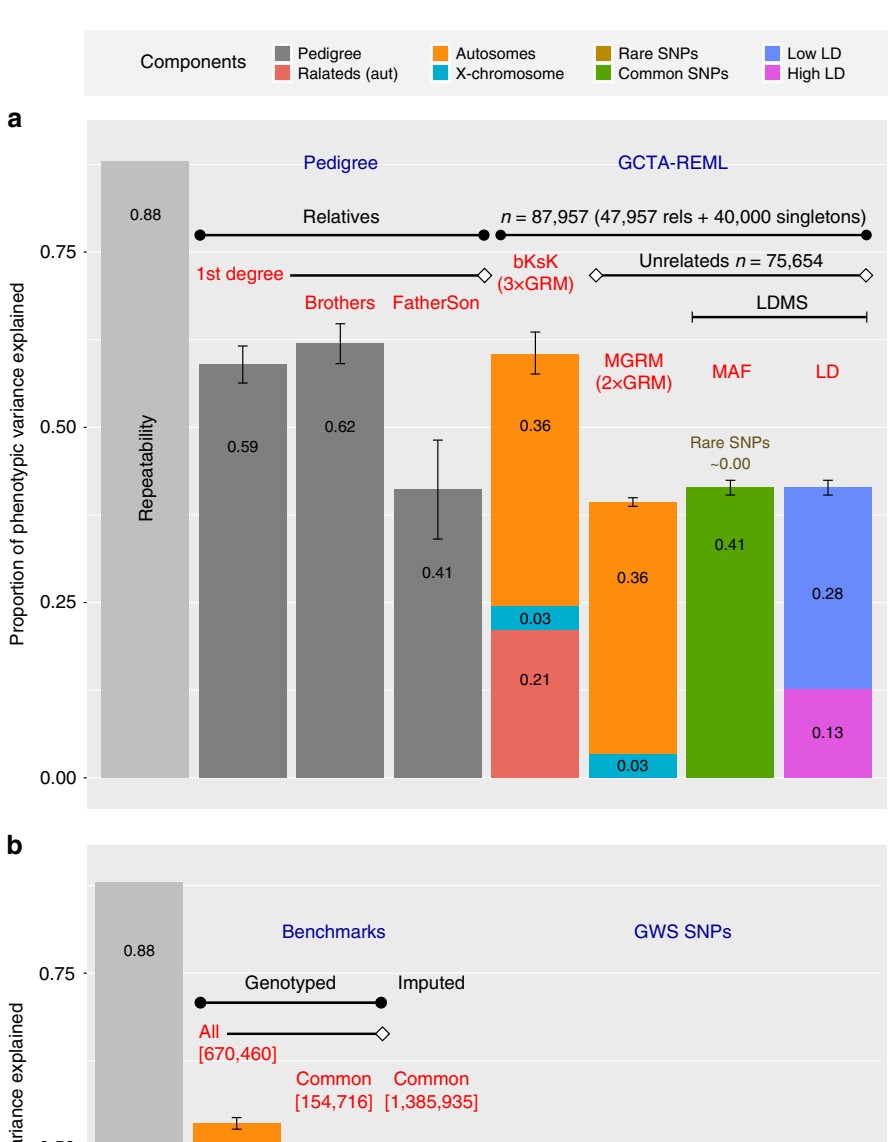

SE = 0.005), 0.211 (SE = 0.030) from the pedigree not captured by common SNPs and 0.033 (SE = 0.002) from the X-chromosome (Fig. 2a, Supplementary Data 1).

Using only unrelated individuals ($n$ = 75,654) and the autosomes, we then added rare SNPs (1e-3 < MAF ≤ 0.01) and stratified the variants across MAF and LD bins using GCTA GREML-LDMS[23]. The analysis accounts for uneven distribution of variance explained by SNPs with different MAF and LD scores[23]. The total $h^2_{SNP,LDMS}$ estimate was 0.415 (SE = 0.028, 25,246,483 autosomal SNPs, Supplementary Data 1). Genetic variation for MPB was enriched among common (0.01 < MAF ≤

0.5) variants ($h^2_{SNP,LDMS,common}$ = 0.434, SE = 0.008, fold-enrichment = 3.63, $P$ = 3.6e-177, see equation (4) for Z-score calculation) and in lower LD regions ($h^2_{SNP,LDMS,lowLD}$ = 0.283, SE = 0.027, fold-enrichment = 1.36, $P$ = 8.7e-3) (Fig. 2a, Supplementary Data 1). Note that this analysis was performed without constraint, and $h^2_{SNP,LDMS,common}$ exceeded the overall $h^2_{SNP,LDMS}$ estimate as $h^2_{SNP,LDMS,rare}$ was negative. A description of the statistical test for enrichment is provided in Methods. Supplementary Data 2 provides results on a finer grid of 12 LDMS components: three MAF bins (1.5e-5 < MAF ≤ 1e-3; 1e-3 < MAF ≤ 0.01; 0.01 < MAF < 0.5), and four LD score quartiles,

**Fig. 2** $h^2$ and variance estimations in the UKB dataset using GCTA-GREML[20, 21]. Contributions of each component are superimposed upon the corresponding bar. Error bars denote the standard error. A corresponding table of values is provided in Supplementary Data 1. **a** Comparisons of $h^2$ estimates. The light grey 'Repeatability' bar represents the upper limit of variance explained by genetic factors. Darker grey bars denote pedigree estimates ($h^2_{ped}$) from pairs of 1st degree relatives ($n = 9449$ individuals or 5217 pairs), brothers ($n = 8010$ individuals, or 4428 pairs) and father–son relationships ($n = 1482$ individuals, or 789 pairs). Coloured bars represent results from partitioned $h^2$ analyses. 'bKsK' bar denotes big K small K $h^2$ estimates fitting three GRMs: (1) standard GRM using autosomal SNPs to calculate relatedness, (2) big K GRM such that rel < 0.05 was set to 0 and (3) using X-chromosome SNPs to calculate relatedness. 'MGRM' shows $h^2_{SNP}$ estimates after simultaneously fitting GRMs based on common autosomal and X-chromosome SNPs, using a dataset only containing classically-unrelated individuals (rel < 0.05). 'LDMS (MAF)' shows GREML-LDMS[23] results, compartmentalising by MAF (rare SNPs defined as 1.5e-5 < MAF ≤ 0.01, versus common SNPs: MAF > 0.01). Note that the contribution of rare SNPs was effectively zero (Supplementary Data 2). 'LDMS (LD)' illustrates compartmentalisation by low versus high LD (cut-offs determined using genome-wide LD scores from individual SNPs). **b** Comparison of variance estimates calculated using COJO-selected GWS SNPs to build the GRM. $h^2_{SNP}$ estimates from GRMs built using (i) genotyped or imputed SNPs and (ii) common or all variants are provided as benchmarks. The number of SNPs used for each analysis (autosomes + X-chromosome SNPs) is provided in red text and square brackets

and further details on the genetic markers used for each of the components. Common variants contributed essentially all $h^2_{SNP}$, despite only accounting for 29% of SNPs in the GREML-LDMS analysis. $h^2_{SNP}$ gradually declined across LD score quartiles: quartiles 1 to 4 contributed 35%, 34%, 19% and 13% of $h^2_{SNP}$, respectively (Supplementary Data 2).

In summary, the results from our variance component analyses are consistent with $h^2 \approx 60\%$, including a contribution of 3–4% from the common SNPs located on the X-chromosome, with almost 60% of additive genetic variance captured by common SNPs, and with an enrichment of genetic variance among common SNPs and lower LD regions (Fig. 2a, Supplementary Data 1).

**Detection of MPB-associated variants**. Our GWAS identified 50,571 autosomal genome-wide significant (GWS) SNPs ($P < 5e-8$ from the BOLT-LMM[24] infinitesimal mixed-model association test, Fig. 3b, Supplementary Data 3), and 8126 X-chromosome loci ($P < 5e-8$, Fig. 3c, Supplementary Data 3). We used GCTA-COJO[25] to identify conditionally-independent SNPs using a joint model, conducting separate analyses for autosomal and X-chromosome summary statistics. This gave 624 conditionally-independent loci: 598 autosomal (Supplementary Data 4) and 26 on the X-chromosome (Fig. 3c, d, Supplementary Data 5). The 624 conditionally-independent SNPs explained 0.252 (SE = 0.011) of MPB variance, with 0.223 (SE = 0.010) from the autosomal subset and 0.029 (SE = 0.008) from the X-chromosome subset (Fig. 2b, Supplementary Data 1). Some X-chromosome SNPs had extremely large effect size, so some variants in high LD remained after COJO analysis (Supplementary Data 5, Fig. 3d). We performed selective multiple regression on these SNPs, eliminating a further two X-chromosome SNPs (Supplementary Data 6). Ultimately, the net product was 622 loci: 598 autosomal and 24 on the X-chromosome (Fig. 3, Supplementary Data 4−6).

**Downstream analyses**. We performed downstream analyses of the GWAS summary statistics to better understand the genetic architecture of MPB, and potential functional consequences.

The FUMA SNP2GENE analysis[26] provided counts of GWS SNPs that were represented in various ANNOVAR annotated categories on the autosomes. The vast majority of SNPS were located in intergenic and intronic regions, with limited representation from 3′-UTR and 5′-UTR, downstream, upstream and exonic regions (Supplementary Fig. 3a).

We also performed partitioned $h^2$ analysis by cellular functional annotation, implemented in LDSC[27] on autosomes. The following categories were significantly enriched for $h^2_{SNP}$ after multiple-testing correction: conserved regions ($P = 7.6e-8$),

histone modifications (H3K27ac, H3K4me1, H3K4me3, H3K9ac; $1.2e-17 < P < 1.7e-7$), super enhancers ($P = 3.5e-11$), DNAse hypersensitivity sites (DHS; $P = 2.8e-8$), digital genomic footprint (DGF; $P = 5.7e-7$) and transcription factor binding sites (TFBS; $P = 1.61e-6$) (Supplementary Fig. 3b, Supplementary Data 12). We also performed partitioned $h^2$ analysis by cell-type group (discussed in Supplementary Note 1).

We used FUMA MAGMA gene analysis[26] to investigate how the MPB GWAS results relate to gene function and expression, finding 850 autosomal genes that survived Bonferroni correction (Supplementary Data 13). Subsequent competitive gene-set analysis demonstrated enrichment for mesenchymal lineages (skeletal, cartilage, and appendage elements), skin and epidermal development, and transcriptional elements (Supplementary Data 14). FUMA GENE2FUNC[26] differentially expressed genes analysis (using GTEx v6) demonstrated enrichment for expression in skin and vaginal tissue among these genes (Supplementary Figs. 4, 5).

We uploaded the 624 COJO SNPs to ICSNPathway[28], which prioritises SNPs with functional annotations to identify associated genes, then groups genes into pathways. This identified 85 candidate causal pathways with enrichment, including the ontologies 'Reproductive' and 'Reproductive Pathways'. These results are discussed in Supplementary Note 1.

**Pleiotropy**. We used LDSC[29] to search for evidence of pleiotropy in the UKB. Of eight female-limited traits (Supplementary Data 7), one genetic correlation ($r_g$) with MPB was significant (Bonferroni-corrected threshold $P < 0.006$): age of menarche ($r_g = -0.09$, SE = 0.02, $P = 2.2e-8$) (Table 1). Among male-limited traits, $r_g$ between age of facial hair onset and MPB was strongest ($r_g = -0.18$, SE = 0.03, $P = 1.7e-11$), and $r_g$ for age at voice breaking was also significant ($r_g = -0.11$, SE = 0.03, $P = 1.6e-5$). Taken together, the negative $r_g$ with ages at facial hair onset, voice breaking and menarche (in females), suggest genetic covariance between earlier puberty onset in both sexes and increased MPB severity (Table 1). Male traits demonstrated stronger correlation than female traits (Table 1). As a quality check, all three traits had $h^2_{SNP} > 0.05$ (estimated by LDSC, Supplementary Data 8).

We used LDHub[30] to estimate $r_g$ between MPB and over 800 other traits (thresholding for $h^2_{SNP} < 0.05$ and FDR-adjusted $P < 0.05$), finding additional statistically-significant $r_g$ between MPB and bone mineral density (BMD; both the lumbar spine BMD studies from 2012[31] ($r_g = 0.12$, SE = 0.04) and 2015[32] ($r_g = 0.15$, SE = 0.04), as well as the UKB heel BMD[30] ($r_g = 0.07$, SE = 0.02)), age of menarche 2014 GWAS[33] ($r_g = -0.09$, SE = 0.02), difference in height between childhood and adulthood at age 8[34]

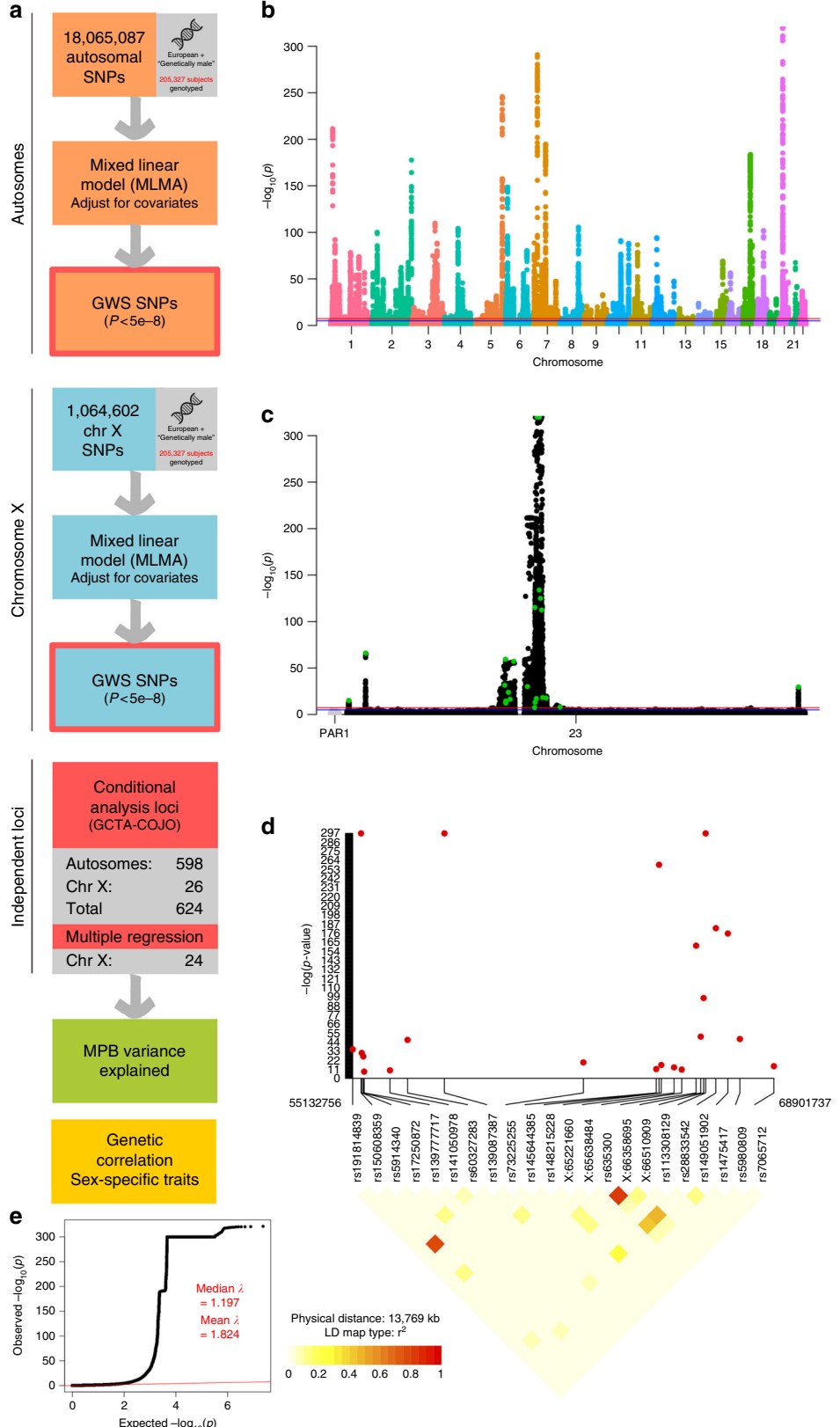

**Fig. 3** MPB GWAS. **a** Workflow diagram. **b**, **c** Manhattan plots for the autosomal GWAS (**b**) and the XWAS (**c**), depicting *p*-values from the BOLT-LMM[24] infinitesimal mixed-model association test. Green points denote 26 GWS SNPs identified by the COJO[25] analysis. Chromosome 23 is the X-chromosome and chromosome PAR1 is pseudo-autosomal region 1. **d** LD plot for the p22.31-q13.2 region, containing jointly significant X-chromosome SNPs from the COJO[25] analysis. **e** QQ-plot for the aggregated set of autosomal and X-chromosome GWAS *P*-values

**Table 1 Genetic correlations of MPB**

| Trait | $N_{individuals}$ | $n_{SNPs}$ | $h^2$ (LDSC) | $r_g$ (SE) | P |
|---|---|---|---|---|---|
| *UKB: Sex-limited traits* | | | | | |
| Birth weight (first child) | 149,365 | 1,012,985 | 0.103 | −0.03 (0.02) | 0.14 |
| First birth (age of) | 126,958 | 1,012,966 | 0.168 | 0.04 (0.02) | 0.05 |
| Last birth (age of) | 126,793 | 1,012,966 | 0.094 | 0.02 (0.02) | 0.37 |
| Live births (no.) | 188,170 | 1,012,989 | 0.062 | −0.06 (0.02) | 0.01 |
| Menarche (age of) | 182,937 | 1,012,994 | 0.259 | −0.09 (0.02) | 2.2e-08 |
| Menopause (age of) | 107,722 | 1,012,954 | 0.133 | 0.01 (0.02) | 0.68 |
| Menstrual cycle (days) | 32,552 | 1,012,843 | 0.007 | 0.03 (0.17) | 0.84 |
| Stillbirths (no.) | 59,258 | 1,012,843 | 0.010 | 0.07 (0.11) | 0.53 |
| Prostate cancer | 6381 cases; 202,043 controls | 1,013,066 | 0.026 | −0.01 (0.03) | 0.84 |
| Facial hair (male, relative age) | 154,439 | 1,013,036 | 0.122 | −0.18 (0.03) | 1.7e-11 |
| Voice broke (male, relative age) | 147,919 | 1,013,026 | 0.074 | −0.11 (0.03) | 1.6e-05 |
| *LDHub: Puberty traits* | | | | | |
| Menarche (age of) (Perry) | 182,416 | 2,441,815 | 0.204 | −0.09 (0.02) | 1.0e-04 |
| Diff in height (age 8 vs adult) | 18,737 | ~2,500,000 | 0.357 | −0.16 (0.04) | 1.0e-04 |
| *LDHub: Other traits* | | | | | |
| BMD: Lumbar spine (2015) | 53,236 | 4,268,111 | 0.128 | 0.15 (0.04) | 3.0e-05 |
| BMD: Heel T-score (UKB) | 194,398 | 10,894,597 | 0.296 | 0.07 (0.02) | 2.0e-04 |
| HOMA-B | 36,466 | ~2,500,000 | 0.087 | 0.12 (0.04) | 1.7e-03 |

Genetic correlations ($r_g$) between MPB and sex-limited traits from the UK Biobank dataset, as well as statistically-significant ($P < 0.05$, FDR-adjusted) traits from LDHub[30]. Analyses were performed using LDSC[27] software, and drew upon autosomal data only

($r_g = -0.16$, SE = 0.04), and HOMA-B[35] ($r_g = 0.12$, SE = 0.04) (Table 1, Supplementary Data 8). LDHub reproduced $r_g$ that we had identified in UKB GWAS, with ages at facial hair onset ($r_g = -0.23$, SE = 0.04, $P = 2.2e-8$), voice breaking ($r_g = -0.12$, SE = 0.04, $P = 6.0e-4$) and menarche ($r_g = -0.09$, SE = 0.02, $P = 1.0e-4$) (Supplementary Data 8).

The trait 'number of children fathered' had a low $h^2$ from the LDSC analysis, so we calculated the phenotypic correlation with age-adjusted MPB. The correlation was weak but significant (Pearson's $r = -0.026$, 95%CI: $-0.030$ to $-0.021$, $n = 203,838$, two-sided $P < 2e-16$). The regression coefficient of the number of children fathered against age-adjusted MPB was $-0.030$ (SE = 0.003, $n = 203,838$, $P < 2e-16$), which corresponds to 0.09 fewer children between men with MPB score of 4 versus 1. This also equates to $-0.033$ children per standard deviation of MPB. Given that the mean number of children fathered is 1.81, and assuming that relative lifetime reproductive success (number of children standardised by the mean) is a proxy for fitness, we infer a linear selection gradient of $-0.018$ per one standard deviation increase in MPB[36].

## Discussion

MPB is a complex trait with interesting genetic characteristics, and a primary focus of our study was to explore the contribution of genetic factors to its variance. Our pedigree-based $h^2$ analysis exploiting first-degree relationship pairs in the UKB gave estimates as high as 0.62 (Fig. 2a, Supplementary Data 1). In comparison, $h^2_{SNP}$ was 0.393 based on common autosomal and X-chromosome SNPs, and was 0.415 with inclusion of rare SNPs ($1.5e-5 < MAF \le 1e-3$). The lower father–son correlation compared to brother–brother pairs (Fig. 2a) may partially reflect X-chromosome contributions, since fathers and sons do not share X-chromosome loci by descent. Given the considerable X-chromosome influence on MPB, we suggest that the brother–brother estimate may give a better estimate. The remaining disparity ($h^2_{ped}$ difference = 0.208, $P = 0.006$ from a two-sided test) may reflect sampling variance, non-additive effects, and common environment shared by brothers. Our big K small K analyses also showed that the pedigree-based estimate

captured X-chromosome $h^2_{SNP}$ as the two-GRM and three-GRM (with X-chromosome) estimates were both around 0.60 (Supplementary Data 1).

MPB $h^2$ has previously been reported to be 0.81 from a twin study using self-reported MPB data on a 13-category scale[7]. However, this estimate is not directly comparable to our pedigree estimate of MPB ($h^2_{ped} = 0.62$) due to differences in trait measurement and analysis method. In addition to sampling variation, the difference in measurement scale and therefore measurement error is probably the main source of differences between the estimates. Gianola (1979)[37] derived the heritability on the observed scale ($h^2_{o[o]}$) for any linear combination of scores in $n$ ordinal categories, when the heritability on the scale of liability is $h^2_l$. In the case of observations on MPB, we assume an underlying continuous scale of liability, $n = 4$ and observations $Y_0$ takes on values 1–4 with frequencies $\pi_j$. This gives Eq. (1):

$$E\left[h^2_{o[o]}\right] = \frac{h^2_l \left(\sum_{j=1}^{n-1} z_j\right)^2}{V(Y_o)} \quad (1)$$

where $z_j$ the height of the normal curve corresponding to threshold $j$ (4 categories gives 3 thresholds), and $V(Y_o) = \sum_{j=1}^{n} \pi_j w_j^2 - \left(\sum_{j=1}^{n} \pi_j w_j\right)^2$, with $w_j$ taking on values 1, 2, 3 and 4. Therefore, it follows that:

$$h^2_l = \frac{h^2_{o[o]} V(Y_o)}{(z_1 + z_2 + z_3)^2} \quad (2)$$

Using Eq. (2) to transform our estimates on the observed 1 to 4 scale to the scale of liability increases the estimates by a factor of approximately 1.18. Hence, assuming a continuous scale of liability, our transformed estimates of total $h^2$ is $1.18 \times 0.62 = 0.73$.

A recent MPB genetic analysis using the UKB and other datasets reported a very high $h^2$ of liability to baldness of 1 and claimed that this was driven by common variants[16]. The authors created a binary score from UKBv1 data, defining 'cases' (affected) as individuals with scores 3 and 4 and 'controls' (unaffected) as those with a score of 1. They then transformed the $h^2$ estimate

on this 0–1 scale to that of liability, using the proportion of affected individuals (i.e. those with a score of 3 or 4) in the sample (i.e. those with a score of 1, 3 or 4) as the population prevalence. However, excluding individuals with a score of 2 creates an upward bias, and this ascertainment is unaccounted for. If liability to MPB is linear in the scores 1 to 4 (as we have found[38]), then the reported estimate of 1 by Pirastu et al.[16] is biased by about one-third, so the actual estimate is consistent with our results.

Our estimate of the proportion of phenotypic variance captured by common SNPs is comparable to other $h^2_{SNP}$ estimates for traits such as height[39] and childhood intelligence[40]. Our GREML-LDMS analysis on the autosomes showed that common SNPs (MAF > 0.01) essentially explained all $h^2_{SNP}$ (fold-enrichment = 3.63, $P = 3.6e-177$), whereas rare SNPs (1.5e-5 < MAF ≤ 0.01) made a very minor contribution. There was also enrichment in lower LD regions (68%, fold-enrichment = 1.36, $P = 8.7e-3$) (Fig. 2a, Supplementary Data 2). However, this analysis did not capture X-chromosome variants, which is relevant as the MPB-associated $AR$ region is one of the highest LD regions on the X-chromosome[11]. The enrichment for common autosomal variants in lower LD regions reflects a recent BayesS analysis on the UKBv1 dataset, which found a signature of negative selection among MPB-associated loci, with an S-parameter estimate of −0.346[41]. The dominating contribution of common variants to MPB was also observed among the 624 conditionally-independent GWS SNPs (autosomes and X-chromosome). Altogether, these SNPs explained 25% (SE = 1%) of phenotypic variance (Fig. 2b) and 98% ($n = 612$) were common (MAF > 0.01) (Supplementary Data 4, Supplementary Data 5). Hence, even accounting for rare variants, the missing $h^2$ is ≈0.2, which may suggest environmental or non-additive inflation of the estimate.

A BayesS analysis showed that MPB is less polygenic than classical complex traits such as height, educational attainment and blood pressure[41]. Compared to the 2014 height GWAS ($n = 253,288$)[39], our MPB analysis has comparable sample size, and we detect a similar number of autosomal associated loci (MPB: 598 versus height: 697). However, the top MPB loci tend to have larger effect size, and explain more variance than the top height loci (Supplementary Note 2, Supplementary Fig. 6).

We performed the largest GWAS to date, exceeding previous studies by over 130,000 individuals. We used a mixed-model GWAS over conventional SNP-by-SNP analyses, allowing us to include relatives and increase power[24]. Compared to Hagenaars et al.[8] (who analysed UKBv1), we replicated 219/287 SNPs within our GWS SNPs (Supplementary Data 9). Pirastu et al.[16] found 107 conditionally independent SNPs (103 autosomal SNPs and four X-chromosome, corresponding to 71 loci) from a UKBv1 case-control design, that they had replicated in separate cohorts at $P < 0.05$. Of these, we replicated 71/103 autosomal and all four X-chromosome SNPs, corresponding to 61/71 loci (Supplementary Data 10). Prior to 2017, 12 GWS MPB-associated loci had been identified, representing 16 independent SNPs[18], which were all replicated ($P < 5e-8$) in our GWS results (Supplementary Data 11).

Our variance analyses confirmed that the X-chromosome disproportionally contributes to MPB, relative to its physical length and number of genes. The X-chromosome makes up around 5% of the human genome in length, and contains 800–900 of the 20,000–25,000 genes in humans (4.5% at most). Here, 624 conditionally independent GWS SNPs accounted for $h^2_{SNP,COJO} = 0.252$ (SE = 0.011), of which X-chromosome conditionally-independent GWS SNPs explained 0.029 (SE = 0.008), or 11.6% (0.029/0.252). In comparison, all common X-chromosome variants explained 8.8% (0.034/0.393) of $h^2_{SNP}$ due to common variation (Supplementary Data 1). This suggests that MPB genetic

architecture on the X-chromosome is less polygenic than on the autosomes, which is reflected by the large-effect X-chromosome loci (Fig. 3c, Supplementary Data 5). We discuss X-chromosome loci annotations further in Supplementary Note 1.

Downstream analyses highlighted interesting and plausible biological pathways that may provide clues for molecular analyses. These are discussed in detail in Supplementary Note 1.

As MPB has a androgenetic basis, and because there is sexual dimorphism in androgenetic regulation[42], we looked for relationships with other sex-limited traits. We analysed eight female-specific traits with a large sample size (Table 1) in search of proxy traits that occur when females carry multiple MPB risk alleles. MPB showed significant $r_g$ in females with age of menarche ($r_g = -0.09$, SE = 0.02, $P = 2.2e-8$), after Bonferroni correction ($P < 0.006$). In males, we observed significant $r_g$ with earlier facial hair appearance ($r_g = -0.18$, SE = 0.03, $P = 1.7e-11$) and earlier voice breaking ($r_g = -0.11$, SE = 0.03, $P = 1.6e-5$). These results indicate a novel association between MPB and earlier onset of puberty in both sexes. These three traits had $h^2_{SNP} > 0.07$ (estimated by LDSC), which is reassuring as $r_g$ can be hard to interpret when $h^2_{SNP}$ is small (Table 1, Supplementary Data 8). LDHub strengthened evidence for association between MPB severity and earlier puberty via earlier age of menarche in the independent Perry et al.[33] dataset ($r_g = -0.09$, SE = 0.02, $P = 1.0e-4$), and reduced difference in height between childhood (aged 8) and adulthood[34] ($r_g = -0.16$, SE = 0.04, $P = 1.0e-4$) (Table 1). Although the X-chromosome (and X-linked androgen pathway components) is excluded from LDSC analyses, these results may be detecting autosomal androgen pathway components (such as $HDAC4$, $HDAC9$, $FOXA2$, $TWIST1$ and $TWIST2$)[18]. Hence, $r_g$ may be underestimated if the genetic basis of earlier puberty onset also has X-chromosome components.

The association between early-onset puberty and hair biology may be reconciled. Androgens regulate hair growth, and simultaneously promote facial hair while causing scalp hair regression in older men[43]. Although the testes are the major source of androgens in males from puberty, the adrenal glands are an important source in both sexes during the earliest detectable stage of sexual maturation: adrenarche. Adrenarche occurs around the age of six, precedes pubarche (pubic hair development)[44], correlates with puberty onset in both sexes[45], and may initiate puberty via hypothalamic-pituitary-gonadal axis activation[46]. Excess adrenal androgens may induce precocious puberty, manifested in congenital adrenal hyperplasia (CAH)[46]. Both androgen sources converge upon the androgen signalling pathway, in which the X-linked $AR$ gene is central.

Adrenarche may have a short evolutionary history as it only occurs in humans and higher primates[46], which is consistent with the X-linked $AR$ locus having recently been under selection[11]. Interestingly, mammalian patterned baldness may also have a recent evolutionary history as it is confined to a few primate species[47]. However, the autosomal LDSC partitioned $h^2_{SNP}$ analysis found enrichment among conserved regions of the genome (Supplementary Data 12), which may include the Wnt/β-catenin signalling pathway[48].

LDHub also showed statistically-significant positive $r_g$ between MPB and bone mineral density (BMD) of lumbar spine ($r_g = 0.15$, SE = 0.04, $P = 3.2e-5$) and heel ($r_g = 0.07$, SE = 0.02, $P = 2.0e-4$) (Table 1, Supplementary Data 8). Early puberty correlates with increased BMD[49], suggesting pleiotropy between all three traits. Our downstream analyses also found enrichment for genes in bone and mesenchymal pathways and lineages. Both MPB and BMD are ageing-related traits with predilection for one sex (osteoporosis is more prevalent in post-menopausal females). Similarly to early-onset puberty, androgen production may underpin both MPB and BMD[50]. This is supported by multiple

lines of evidence: androgen-deprivation therapy for prostate cancer decreases BMD[51], whereas hypogonadal men have low BMD that normalises with testosterone therapy[52]. Further, CAH patients with higher adrenal androgens (such as DHEAS), tend to have higher BMD[53], as do women with high testosterone levels due to polycystic ovarian syndrome (PCOS)[54]. Interestingly, strontium ranelate (an osteoporosis therapy) induces alopecia in some patients[55].

We also observed significant $r_g$ between MPB and HOMA-B (indicating increased pancreatic β-cell function) (Table 1). There are epidemiological associations between early-onset MPB and biomarkers for metabolic dysfunction such as HOMA-IR[56] (insulin resistance), but not HOMA-B. Interestingly, genetic loci contributing to the difference in height between childhood and adulthood (a marker for early puberty) have been associated with age of menarche as well as adiposity[34] (also a marker of metabolic dysfunction). The association of MPB with all of these traits adds to the case for pleiotropy. This is biologically plausible as adipose tissue can store androgens and metabolise them into oestrogens. Further, testosterone may increase metabolic dysfunction such as insulin resistance by decreasing the hormone adiponectin[57].

LDHub analyses did not find $r_g$ between MPB and previously described epidemiological associations (for prostate cancer, and coronary artery disease), and the $h^2_{SNP}$ estimate for type 2 diabetes in this analysis was too low for the estimate to be reliable ($h^2_{SNP} = 0.036$, SE = 0.003).

Overall, a proposed model to integrate our genetic correlation results is thus: MPB is driven by the androgen pathway, with components on both autosomes and the X-chromosome. Androgens are produced by adrenals (in both sexes) and gonads (particularly testes, in men). Androgen signalling is associated with puberty onset, which explains $r_g$ between MPB and facial hair appearance, voice breaking, age of menarche and difference in height between childhood (aged 8) and adulthood (Table 1). The $r_g$ between early-onset puberty and BMD also manifests clinically, and BMD is subject to androgen modulation. Hence, the $r_g$ between MPB and BMD may plausibly be due to androgen-related pleiotropy. The correlation with HOMA-B is also supported by epidemiological links with metabolic disease risk factors, reduced height difference between childhood and adulthood, and the interrelationship between androgen metabolism and adipose. Again, these analyses were only based on autosomal data, which excludes the X-chromosome contribution to MPB. There may also be gene-level evidence for pleiotropy among the GWS COJO SNPs (Supplementary Note 1).

The genetic correlation analyses may also show evidence of weak negative selection in the contemporary UK population or alternatively, pleiotropic selection. Our search was motivated by previous associations of altered androgen regulation with infertility[58] and prior identification of MPB susceptibility loci within haplotypes associated with reduced fertility[3]. In males, we took the number of children fathered as a male fertility trait, finding negative phenotypic correlation with the number of children fathered (Pearson's $r = -0.026$, 95%CI: $-0.030$ to $-0.021$). More concretely, men with MPB score of 4 on average had 0.09 fewer children than those with MPB score of 1, after adjusting for age. Furthermore, the significant linear selection gradient of $-0.018$ ($P < 2.2e\text{-}16$) in the contemporary UK population is similar in magnitude to that reported for other traits in the UKB[36]. With regards to MPB and natural selection, it has been suggested that MPB is a sign of social maturity (independent of age) and increased appeasement, which may compensate for decreased attractiveness in reproductive success[59]. Genetically, the jointly-independent autosomal SNP rs199441 (Supplementary Data 4) is within the 17q21.31 H1 haplotype which is under negative

selection among Europeans[3], whereas the AR/EDA2R haplotype (containing the major MPB locus) has recently undergone positive selection[60].

Among female-specific fertility traits, LDSC identified nominally significant ($P < 0.05$) $r_g$ between increased MPB severity and fewer number of live births ($r_g = -0.06$, SE = 0.02) and increased age of first birth ($r_g = 0.04$, SE = 0.02) (Table 1), suggesting shared genetic risk. These directions of effect point towards lower female fertility, and negative selection. There is also S-parameter evidence for negative selection on age of menarche (which is permissive for reproduction), and on age at first live birth[41]. These two traits show opposing directions of effect between (a) lifetime reproductive success (LRS) (age of menarche: positive; age at first live birth: negative[36]) and (b) $r_g$ with MPB (age of menarche: negative; age at first live birth: positive), which is consistent with MPB being negatively correlated with LRS and fertility. The enrichment for low LD variants in our $h^2$ analyses support a case for weak negative selection in MPB, as do downstream analyses indicating relationships between MPB COJO SNPs and fertility-related genes (Supplementary Note 1).

Given our results linking MPB to BMD, MPB may be a useful model trait for other late-onset diseases that are biased towards one sex (such as osteoporosis[61] and sarcopenia[62]). Biologically, such traits share a basis in sex-hormone regulation, so it will be interesting to investigate pleiotropy and similarities in genetic architecture.

In summary, we show that MPB is a highly polygenic, partially X-linked and heritable trait in the UKB, and that almost 60% of the additive genetic variation is captured by common SNPs. We find plausible genetic evidence to suggest androgen-related pleiotropy with early-onset puberty, BMD and metabolic traits, and find that MPB is an interesting genetic trait in its own right. Overall, our results for MPB are consistent with the emerging genetic and evolutionary properties of many complex traits in human populations[63]. More broadly, the availability of extremely large datasets of genome-wide genotypes and phenotypes in human populations allows insights into the genetic architecture of complex traits, and the pleiotropic mechanisms that may underlie proxy traits or medical comorbidities.

## Methods

**Phenotype data**. The UKB dataset contains MPB self-report data from 224,897 males. Individuals were asked to match their balding pattern to four images (Supplementary Fig. 1). MPB scores 1 to 4 respectively correspond to no balding, vertex balding, crown balding and a combination of vertex and crown balding. Of the 224,897 males, we selected 205,327 who self-identified as British, Irish or other White ancestry, were genetically-European from the leading principal components (PCs) of the SNP genotype data, had genotyping data available from the UKB Phase 2 (UKBv2) genotyping and imputation release, and were confirmed to be genetically-male. Our phenotypes for analysis are residuals from regressing the MPB scores on age, assessment centre, ethnicity and 40 principal components calculated using the UKB European population (Fig. 1d). Throughout the manuscript, the residuals are referred to as 'adjusted MPB scores' (Fig. 1d).

**Genotype data**. Supplementary Fig. 2 provides an overview of the data quality control (QC) workflow used across analyses, performed using PLINK 1.9[64,65]. We used autosomal genotype data imputed to the Haplotype Reference Consortium (HRC) reference panel provided by the UKB[66]. The UKB-provided genotype probabilities were used to hard-call the genotypes for variants with an imputation info score > 0.3. We used the hard-call-threshold 0.1, setting the genotypes with $P \leq 0.9$ as missing. For the autosomal GWAS analysis, we retained the hard-called variants with minor allele frequency (MAF) > 0.0001, missing genotype rate < 0.05 and Hardy–Weinberg equilibrium (HWE) $P > 1e\text{-}6$ as computed in the sample of unrelated Europeans, such that 18,065,087 autosomal SNPs remained. For the heritability analyses, 1,123,347 HapMap3 variants passed our quality control filters (MAF > 0.01, missing genotype rate < 0.05 and HWE $P > 1e\text{-}6$) in the set of 456,419 male and female Europeans; these were used to estimate a GRM for 205,327 males.

Imputed X-chromosome genotypes were not provided in the full UKB release from 2017. We imputed the non-pseudo-autosomal and pseudo-autosomal region (PAR1) to the 1000 Genomes phase 3 mixed population reference panel using IMPUTE2. The imputation pipeline followed is available at https://github.com/CNSGenomics/impute-pipe. The genotyped SNPs used for imputation were those used for X-chromosome UKB phasing input, passing additional QC filters (the common set of SNPs with missing genotype rate < 0.05 and MAF > 0.0001 in both males and females and HWE $P > 1e-6$ in females, as males are haploid) in the set of unrelated Europeans, excluding those participants that were (1) heterozygosity or missingness outliers, (2) had inconsistent reported and biological sex, (3) were aneuploid, (4) if they had been excluded from kinship inference by the UKB or (5) if they had withdrawn their consent. After the imputation, we retained markers with imputation info score > 0.3 and again estimated QC metrics in the set of unrelated Europeans. For the non-pseudo-autosomal region, we kept the markers with MAF > 0.0001 in both males and females, and those with HWE $P > 1e-6$. The same thresholds applied for PAR1, without stratifying by sex. This left 1,064,602 X-chromosome SNPs (1,024,430 non-PAR and 40,172 PAR1 SNPs) for use in GWAS. For the heritability analyses, we used a subset of 262,588 X-chromosome markers with MAF > 0.01, estimated in the 205,327 males.

In addition, we created an LD pruned (LD $r^2 > 0.9$, window size = 1000 SNPs, step size = 100) set of common autosomal HapMap3 variants (MAF > 0.01) to calibrate the GWAS analysis, as required by the BOLT-LMM software[24].

**Ethics**. The UK Biobank study protocol has ethics approval from the North West Multi-centre Research Ethics Committee (covering the UK), the Research Tissue Bank, and all participants provided written informed consent (https://www.ukbiobank.ac.uk/the-ethics-and-governance-council/).

**Female data**. We used data from females in our genetic correlation analyses for the following female-limited traits: birth weight of first child, age at first child, age at last child, age of menarche, age of menopause, length of menstrual cycle (only including women with regular periods), number of live births and number of stillbirths (Supplementary Data 7). For each of the phenotypes, we selected for analysis genetically-female individuals who were genetically-European from the leading principal components (PCs) of the SNP genotype data, and whom had genotyping data available from the UKBv2 genotyping and imputation release. Table 1 gives the number of female participants in each genetic correlation analysis.

**Narrow-sense heritability ($h^2$) estimation**. To benchmark all estimates of variance explained, we estimated repeatability: the correlation between repeated observations on the same individuals. Repeatability can be due to genetic and permanent environmental effects and is an upper limit of a trait's broad sense heritability ($H^2$)[67]. We estimated repeatability by calculating the correlation between MPB measurements taken from 2006–2010, to those taken from the same individuals during 2012–2013 ($n = 9603$).

UKB-provided kinship estimates[66] were used to infer first-degree relationships. Among MPB males, we identified a set of 9449 individuals (5217 pairs) with a first-degree relative. Of these, a subset of 1482 males were in a father–son relationship (789 pairs), and 8010 males were in a brother–brother relationship (4428 pairs). For each of these three sets of individuals, we estimated variance components using REML to simulate a true pedigree analysis, with first-degree relationships set to 0.5 and all others set to 0, using GCTA software[20,21].

We performed a number of variance component analyses to estimate and partition phenotypic variation associated with common SNPs in both closely-related and conventionally-unrelated individuals. These analyses help dissect the contribution from common versus rare variants and shared environmental effects, since rare variants are not well tagged by common SNPs, and close relatives may share common environmental effects[20]. Given constraints on computational resources, we maximised the number of individuals with relatives (estimated relatedness from SNP data > 0.05) so as to estimate both variance components with good precision. Specifically, the maximum dataset ($n = 87,957$) used in our $h^2$ analyses included 47,957 individuals with relatives in the data, and a random sample of 40,000 singletons (individuals with no relatives) (Supplementary Fig. 2). We estimated variances associated with both close and distant relatives simultaneously using a 'big K small K' analysis[22] of the full set of 87,957 males. The analysis used two GRMs (built using GCTA[20,21]): the big K GRM in which relatedness coefficients < 0.05 based on autosomes were set to 0, and the conventional GRM constructed from common autosomal (1,123,347 SNPs) and X-chromosome markers (262,588 SNPs). This sample of 87,957 contains a maximum set of 75,654 unrelated males, using a rel < 0.05 threshold on a genetic relationship matrix (GRM) calculated using 1,123,347 common autosomal SNPs. We used these individuals in REML analyses to estimate the $h^2_{SNP}$ attributable to common autosomal SNPs alone, common X-chromosome SNPs alone, and both common autosomal and X-chromosome SNPs.

To partition genetic variation according to allele frequency and linkage disequilibrium (LD) score, we performed a GREML-LDMS analysis[23] (with the no constraint setting) on all autosomal markers ($n = 25,246,483$) from the unrelated subset ($n = 75,654$). We divided SNPs into three bins based on MAF ($1.5e-5 < MAF \le 1e-3$; $1e-3 < MAF \le 0.01$; $0.01 < MAF < 0.5$) and four quartiles based on LD

score in a genome-wide manner, giving a total of 12 LDMS components (Supplementary Data 2). Throughout the manuscript however, we compared components for low versus high LD (based on median LD score), and rare ($1.5e-5 < MAF \le 0.01$) versus common ($0.01 < MAF < 0.5$) variants. We used the individual SNP LD score as this has been shown to be superior[68], defined as the sum of the LD $r^2$ scores between all SNPs within a 20 Mb region centred on that SNP.

We tested for $h^2_{SNP}$ enrichment in the GREML-LDMS components for low versus high LD and rare versus common variants. We define $h^2_C$ and $M_C$ as the heritability estimate and the number of SNPs for category C respectively. Then the fold-enrichment can be calculated by the ratio of per SNP $h^2$, as in Eq. (3):

$$h^2_C \times M / (M_C \times h^2) \qquad (3)$$

where $h^2$ is the total heritability and $M$ is the total number of SNPs included in the analysis.

To test the significance of this enrichment, we can construct Eq. (4).

$$Z = \frac{h^2_C}{M_C} - \frac{h^2 - h^2_C}{M - M_C} \qquad (4)$$

The standard error can be estimated using the covariance matrix for LDMS component estimates.

**Identification of MPB-associated variants**. An additive mixed linear model association analysis (implemented in BOLT-LMMv2.3[24]) was performed using the autosomal and X-chromosome genotype data, with adjusted MPB scores as the dependent quantitative trait. BOLT-LMM accounts for genetic relationships between individuals, so we used the full set of MPB males ($n = 205,327$). We used the LD-pruned autosomal SNPs (described in Genotype data section) to calibrate the analysis via the –modelSnps command. We selected genome-wide significant (GWS) SNPs using a significance threshold $P < 5e-8$, under the infinitesimal mixed model. Note that males are haploid for the X-chromosome. Hence, although the actual number of alleles is 0/1, BOLT-LMM encodes males as 0/2. Thus, the per-allele effect size is half the effect size reported in the summary statistics output for X-chromosome SNPs.

Classical LD-clumping to identify GWS loci is based on a user-specified LD $r^2$ threshold. If this threshold is small then truly independent loci may be excluded; however, use of the standard threshold of $r^2 = 0.2$ may retain SNPs that are associated by LD because some effect sizes are so large. Instead, we used GCTA-COJO[25] to identify a smaller set of jointly-significant MPB-associated SNPs. Although COJO is built for autosomal (diploid) analysis, analysis of the X-chromosome separately from autosomes overcomes this issue.

We opted for the summary statistics option as the –cojo-actual-geno method (that uses the same individual-level genotype data as the for the GWAS) is not built to analyse the X-chromosome.

We quantified the variance explained by the COJO-selected independent GWS SNPs, by creating a GRM from these SNPs on the set of 75,654 unrelated males and estimating the variance explained using REML. Since the discovery and test sets are not independent, we expect a (slight) upward bias in this estimate, even if our criteria for inclusion of SNPs are stringent.

**Downstream analyses**. We used three tools to investigate the top GWS SNPs: FUMA GWAS (Functional Mapping and Annotation of Genome Wide Association Studies)[26], partitioned $h^2_{SNP}$ analyses implemented in LDSC[27], and ICSNPathway[28].

We input the GWAS summary statistics into FUMA[26], using p-values from the non-infinitesimal mixed model. We include here results from the SNP2GENE tool (MAGMA v1.6 gene and competitive gene-set analysis, which uses the 1000 Genomes Phase 3 reference; MAGMA Tissue Expression Analysis, which uses GTEx v6 data; and ANNOVAR functional consequences of SNPs on genes), as well as from the GENE2FUNC tool (examining differentially expressed genes by GTEx v6 tissue types). Note that these analyses only consider autosomal genes.

We used LDSC to partition heritability on the autosomes by functional annotation, and for cell-group-type analysis[27]. We used 1000 Genomes Phase 3 functional annotation categories, which included coding, 3′-UTR, 5′-UTR, promoters, enhancers, introns, histone modifications, DNAse I hypersensitivity sites (DHS), protein binding sites (CTCF and transcription factors) and conserved regions (in mammals), as described by Finucane et al.[27]. Ten cell-type-groups were included in the analysis, including adrenal/pancreas, cardiovascular, central nervous system (CNS), connective tissue and bone, gastrointestinal, haematopoietic, kidney, liver, 'other' (which includes adipose nuclei, breast tissue, ovary, penis foreskin and placenta) and skeletal muscle.

We uploaded COJO SNPs to ICSNPathway[28] to look for functional associations of the GWS SNPs. For this analysis, we used default settings (European American HapMap population, $r^2 > 0.8$, LD neighbourhood distance = 200 kb, FDR < 0.05 cut-off for pathway-based analysis), with the exception of the gene window (looking for genes within 100 kb of the uploaded SNPs).

**Relationships between MPB and other traits**. As the androgenetic pathway is a major contributor to MPB[15,18], and androgen signalling is associated with male fertility[58], we investigated whether MPB is related to reproductive success and

hormone-associated traits in both males and females. For genetic correlation analysis, we selected eight female traits, as described in Female data above, and for males, we selected relative age of first facial hair ($n$ = 152,467 had both MPB and facial hair data available), and relative age of voice breaking ($n$ = 146,130 had both MPB and voice breaking data available). Both traits are scored as 1: 'younger than average', 2: 'average', and 3: 'older than average'. We also included prostate cancer in our genetic correlation analyses (cases = 6381, controls = 202,043). Details of coding, exclusion criteria and UK Biobank Field Codes are provided in Supplementary Data 7.

We used the UK Biobank data and HapMap3 SNPs to generate GWAS summary statistics for each of the eight female traits, and three male traits (facial hair, voice breaking and prostate cancer) (GWAS details in Table 1). We used LDSC[29,69] to calculate genetic correlations between these traits and MPB, based on autosomal data only. LD scores were calculated using the European UK Biobank population. LDSC standard errors are calculated using a block jackknife approach, and $p$-values are calculated by jackknifing the regression numerator and applying a $Z$-test[27].

We input the MPB summary statistics into LDHub[30] to look for genetic correlations with many other traits. At the time of writing, LDHub had 832 GWAS available to test against. We only present results with $h^2_{observed}$ > 0.05 to ensure that the associations were robust.

We also determined phenotypic correlations between the male traits, as well as between age-adjusted MPB with the trait 'number of children fathered' ($n$ = 203,838). Individuals who reported fathering > 20 children were excluded from the analysis.

**URLs**. Data was obtained from the UK Biobank at http://www.ukbiobank.ac.uk/. All analyses were performed using publicly available software: PLINK 1.9 at https://www.cog-genomics.org/plink2; GCTA at http://cnsgenomics.com/software/gcta/; BOLT-LMM at https://data.broadinstitute.org/alkesgroup/BOLT-LMM/; FUMA at http://fuma.ctglab.nl/; LDSC at https://github.com/bulik/ldsc; LDHub at http://ldsc.broadinstitute.org/; and ICSNPathway at http://icsnpathway.psych.ac.cn.

## Data availability

The data used to generate the findings of this study are available from the UK Biobank, under project number 12505 (http://www.ukbiobank.ac.uk/). The individual-level data is available upon application to the UK Biobank. The source data underlying Fig. 2 are provided in Supplementary Data 1. Summary statistics on the SNPs with $P$ < 5e-8 in Fig. 3b and Fig. 3c are provided in Supplementary Data 3. The full GWAS summary statistics are available at http://cnsgenomics.com/data.html. A Reporting Summary for this Article is available as a Supplementary Information file.

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

## Acknowledgements

This research was conducted using UK Biobank resources, under project number 12505. The images depicting the MPB patterns were kindly provided by the UK Biobank on an errors and omissions excepted (E&OE) basis. The authors thank the UK Biobank volunteers without whom this research would not have been possible.

## Author contributions

P.M.V., M.R.R. and N.R.W. conceived the study. C.X.Y., J.S., P.M.V., N.R.W. and M.R.R. wrote the manuscript. C.X.Y. performed analyses with assistance and guidance from J.S. and Y.W. Data and software were contributed by J.S., K.E.K., M.R.R. and J.Y. Resources and funding were contributed by P.M.V., N.R.W. and J.Y. All authors critically reviewed and approved the manuscript.

## Additional information

**Competing interests:** The authors declare no competing interests.

