## [Peer Review File · Nature Communications]

Reviewer #1 (Remarks to the Author):

The authors estimate and dissect the genetic component of MPB, as a model trait for humans. The starting point is clear and interesting, the analysis is thorough and detailed, the conclusions are mostly solid and logical. Below are some major and minor comments that I believe could improve the m/s (or at least my understanding of it).

Major comments:

The focus of the paper is not entirely clear to me, it seems to be jumping between standard GWAS, conditional analysis, stratified heritability estimation, combined with genetic correlation analysis [let's admit that is the general GWAS paper recipe]. It seems that the GWAS was only "necessary" to be able to calculate genetic correlation. If this is correct, please explain clearer the aims of the study.

I do not see the point of the PRS correlation analysis on top of LDSC. What was the reason for that? Is it some sort of approximation for a Mendelian randomization causal estimate? Following up on this, given the epidemiological studies implicating possible link with prostate cancer, cardio-metabolic diseases, I miss causal inference (MR) analysis.

What was the relationship between MAF-h² and LDSC-h² on a finer grid (results are only presented for MAF<0.01 vs MAF>=0.01, low LD vs high LD results)? The contribution would be better quantified if authors made it clear what is the actual proportion of markers with such property. In addition, if low LD markers contribute more (proportionally?), would not it invalidate the LDSC approach (and support LDAK)? For robustness, I'd like to see whether the h² estimates are confirmed by a method that is based on the LDAK model. I do not want to impose too much extra work, but this is the most central point of the paper. If it is such a model trait, would be informative to know how these two methods compare and if there are differences why we see them?

Have the authors performed LDSC stratified by functional categories?

While overall pleiotropy (LDSC result) was quite weak, could the authors identify most prominent pleiotropy among the top hits (GW-significant)?

I feel that the bit on selection is quite thinly covered in the manuscript. “It appears that MPB autosomal genetic variance is mostly driven by common variants in regions of low linkage disequilibrium, consistent with weak negative selection on MPB-associated loci” I do not see any strong quantified evidence presented for this claim, but a list of qualitative statements (mostly common variants, low LD [except chrX], (insignificant but) negative genetic correlation with fathered offsprings). I feel this statement being too general and true for almost any complex trait. Could the authors either expand on this angle or tone it down.

Minor comments:

1. “We identified 54,107 autosomal GWS loci” - I believe these are not “loci”, but SNPs.
2. “The trait “number of children fathered” had a weak phenotypic correlation with age-adjusted MPB” – out of how many tested traits?
3. “There is some evidence of interactions between the loci on the autosomes and the X-chromosome. Separate 241 conditional analyses on each of the autosomal and X-chromosome sets of SNPs gave 705 and 35 conditionally-independent SNPs respectively” – I do not see why this would point to interaction. Could not it simply be some correlation between chrX and autosomal markers? (E.g. through uncorrected pop stratification) Having 5 more independent loci when looking at them independently is not really solid evidence; may not even be significant? It is also contradicted by the fact that the bivariate GCTA showed no significant covariance between the chrX and autosomal GRMs when fitted for MBP.
4. The BOLT-LMM analysis was carried out in 205K samples (out of which >150K are certainly unrelated), but the GCTA was done on 75K unrelated samples only. Why?

Reviewer #2 (Remarks to the Author):

Yap et al. Estimation and Dissection of Genetic Variation for Male Pattern Baldness

The authors have used UKB to do a GWAS of MPB, and argue that it is a model trait. They find the pedigree-heritability to be 0.62 and SNP-heritability to be 0.39. They find 735 loci and these tend to have weak negative selection. The authors devote some focus to the X-chromosome, as this is often ignored in GWAS of MPB and other traits. Here it is somewhat simplified due to the fact that MPB is a sex-limited trait.

The authors are well-versed in heritability/GWAS methods (having invented some of them) and there are few issues with the manuscript.

Major comments:

1) It would be very helpful in the results section to define how you've modeled MPB as a trait. The reader has to dig through the paper to identify that you have generated adjusted MPB scores after residualizing for age, centre, ethnicity and PC's.

2) While the authors have used 4 ordered categories to ascertain the concept of MPB, MPB is clearly not 4 simple ordered categories (rather it is a continuously progressive trait that varies by rate of progression across men). Early onset MPB affects a smaller number of men, whereas nearly all men experience some hair loss later in life. This likely reflects different biology. I therefore have a difficult time agreeing with the authors that MPB is a "model trait" in the same way that height is a model trait. Further arguments against this concept include the high degree of variance explained by the X chromosome, due to the large effect size common variants present there. Few traits that I'm aware of have such a disproportionate effect of the X chromosome.

3) It would be interesting to understand if the genetic correlation analyses (LDSC and PRS) correlate with sexual activity. I'm not certain which measures of sexual activity are present in UKB, but this could easily be explored.

Minor comments:

1) It would be helpful to show the definitions of the MPB categories early in the manuscript and in Figure 1.

2) Most readers will not recall differences in metrics of heritability. While the authors have written lovely articles on these topics, it might be helpful to have a small box reminding the reader of the definitions.

3) When the authors state that they have identified 54,107 autosomal GWS loci, might it make more sense to state 54,107 GWS SNPs?

4) I don't think I understand why the number of COJO loci differs when the autosomal and X chr SNPs are analyzed separately and together. Could the authors explain this?

5) It would be helpful if Table 1 contained the number of cases/controls for case/control analyses (i.e. prostate cancer)

Reviewer #3 (Remarks to the Author):

The authors state their aim to study male patterned balding (MPB) as a 'model genetic trait'. They do this by performing the largest GWAS to date for this trait in the large UK Biobank (UKB) study, and intriguingly they describe the GWAS for MPB as "a by-product of our analysis" (Line 200). I presume this explains why they perform only very limited downstream analysis of the GWAS findings. I find this limits some of the claims made here, whereas if the focus is indeed on broader lessons regarding inheritance then those aspects need further consideration and interpretation.

Comments:

1. A key limitation to MPB being a 'model genetic trait' is the lack of agreement in this field regarding trait measurement and analysis, which leads substantial inconsistencies in heritability estimates and GWAS signals. Although UKB ordered the categories 1-4, the difference between categories 2 (vertex) and 3 (crown) are qualitative rather than quantitative. The age distribution in Fig 1 appears to indicate similar distributions for groups 1-2, while groups 3-4 show a separate close distribution. Genetics could shed important light on this issue. Are groups 2 and 3 relatively distinct? Are 1-2 and 3-4 genetically similar groups?

2. The claimed relevance of MPB to reduced fertility (Abstract Line 15) requires much further substantiation. This claim appears to be based only on a weak ($P=0.01$) genetic correlation with number of live children in women and a weak (and surprisingly isolated) phenotypic correlation with "number of children fathered". Genetic and phenotypic associations between MPB and these traits should be described more systematically in both men and women.

3. The relevance of MPB to other traits is also very limited. Table 1 claims to show genetic correlations with female- and male-limited traits. Several of the (presumed, due to lack of clear labelling) 'female-limited traits' are also relevant to males (e.g. age at 1st/last birth, BW of child, number of children). Several other 'epidemiological links' are listed in Para 1 of the main text and should also be tested.

4. The genetic overlap between MPB and puberty timing appears novel. However, the proposed explanation and discussion (Line 244-252) need further consideration. They say that the link is "easily reconciled" but immediately then indicate the opposite by stating that the traits rely on

different sources of androgens. Which tissues are enriched by expression of genes located near to MPB associated SNPs? Is there a causal relationship between these traits?

5. Regarding the key claim of weak negative selection on MPB-associated loci, Fig 2A shows more heritability explained by low LD vs. High LD components of the genome, but to assess whether there is enrichment requires information on the definition, size and numbers of variants in each component. Analyses should quantify the degree and probability of enrichment.

6. They identify 14 'independent signals' in the 2.3 MB Androgen receptor gene region at Xq12. Because the association signal here is so strong it is difficult to establish whether these are truly independent. Fig 3C should focus more precisely on the X chromosome to more clearly display this region and indicate LD with the strongest Xq12 SNP.

7. Line 74-75 "These results may be inflated by common environment effects and non-additive genetic variation". This statement is vague and should be expanded here and commented on further in the discussion. Do they infer that specifically the brother-brother pairs h^2 estimate is inflated by CE? Inclusion of non-additive genetic variation does not 'inflate' heritability, rather its omission in father-son pairs is 'deflation'. Comment on which estimate is nearer the truth?

8. 'Interaction' between autosomal and X-linked loci is intriguing and needs further comment. What do you mean by interaction and how could this impact GCTA models? Is there correlation between autosomal and X-PAR variants for some reason?

NCOMMS-18-12248: Response to Referees

Reviewer #1 (Remarks to the Author):

The authors estimate and dissect the genetic component of MPB, as a model trait for humans. The starting point is clear and interesting, the analysis is thorough and detailed, the conclusions are mostly solid and logical. Below are some major and minor comments that I believe could improve the m/s (or at least my understanding of it).

We thank the reviewer for their detailed and insightful comments on the manuscript. We have addressed each of the points below.

Major comments:

The focus of the paper is not entirely clear to me, it seems to be jumping between standard GWAS, conditional analysis, stratified heritability estimation, combined with genetic correlation analysis [let's admit that is the general GWAS paper recipe]. It seems that the GWAS was only "necessary" to be able to calculate genetic correlation. If this is correct, please explain clearer the aims of the study.

Re: We thank the reviewer for their helpful comments. Our initial goal was to go beyond standard GWAS by focussing more on genetic architecture. However, based on the comments of all reviewers we now include more extensive use of the GWAS summary statistics, further genetic correlation analyses, as well as downstream analyses. We have included these findings throughout the manuscript (Results **pages 6-8**, Discussion **pages 12-15**, Methods **22**, **Supplementary Note 1**).

Specifically, we analysed our GWAS summary data using the FUMA pipeline to prioritise genes that might be functionally related to MPB (**page 7**). Competitive gene-set analysis demonstrated enrichment for mesenchymal lineages, as well as skin and epidermal development (**Supplementary Table 14**). A particularly relevant finding was the involvement of the Wnt/beta-catenin pathway, which links the mesenchymal aspects of MPB to the androgen pathway (**Supplementary Note 1**).

We also performed h^2 partitioning by cellular elements (**page 6**). The cellular elements that were enriched for MPB variants included histone modifications, which is consistent with the dynamic nature of hair physiology and development (**Supplementary Note 1**).

These analyses – in conjunction with an expanded genetic correlation analysis using LDHub – provided interesting biological insights (**pages 13-15**). LDHub identified r_g with early puberty onset, increased bone mineral density, and increased HOMA-B (a biomarker for increased pancreatic beta-cell function). In particular, the relationship with bone mineral density is concordant with the mesenchymal tissue types identified by downstream GWAS analysis.

An analysis of GWS SNPs using ICSNPathway also gave gene-level evidence for pleiotropy among COJO SNPs that was consistent with the genetic correlation results (**Supplementary Note 3**).

I do not see the point of the PRS correlation analysis on top of LDSC. What was the reason for that? Is it some sort of approximation for a Mendelian randomization causal estimate? Following up on this, given the epidemiological studies implicating possible link with prostate cancer, cardio-metabolic diseases, I miss causal inference (MR) analysis.

Re: We thank the reviewer for this comment. We included the PRS correlation analysis as an alternative means to estimate genetic correlation, only based upon the genome-wide significant SNPs. We have decided to exclude this PRS analysis for simplicity.

As per the reviewer's suggestion, we also ran a Mendelian randomization analysis using the GSMR tool to test the putative effects of prostate cancer (UK Biobank, 34 GWS SNPs), coronary heart disease (CHD – the 2015 CARDIoGRAMplusC4D 1000 Genomes-based GWAS, 65 GWS SNPs) and type 2 diabetes (T2D – the 2017 DIAGRAM 1000G GWAS, 128 GWS SNPs) on MPB. For this analysis, we used UKB phase 3 as the LD reference (note that GSMR

accounts for remaining LD between SNPs although the SNPs were selected to be near-independent by a clumping analysis in PLINK), and found that none of these traits were ‘causal’ for MPB, when using a Bonferroni-adjusted threshold. The results are tabulated below. There was overlap between the participants used in the prostate cancer and MPB GWAS, as they both came from the UK Biobank dataset. However, GSMR is appropriate to use here as the phenotypic correlation between the traits was small ($r_p = -0.0037$, 95%CI: -0.0081-0.0006, $p = 0.08978$), meaning that the bias in the GSMR estimate is likely to be small.

Exposure	Outcome	b_{xy}	SE	p-value	#SNP
CHD	MPB	0.0154	0.0089	0.085	20
T2D	MPB	-0.0138	0.019	0.47	9
Prostate cancer	MPB	-0.2904	0.1378	0.035	32

b_{xy} : the estimated causal effect on the exposure on outcome.

On balance, we believe that causal inference analyses are not biologically relevant in this context, given that MPB is not a true “exposure” in the environmental sense. Hence, we have chosen not to include these results given concerns about length of the manuscript. In addition, there may be genetic correlation between the traits which make pleiotropy and causality difficult to distinguish between.

What was the relationship between MAF-h² and LDSC-h² on a finer grid (results are only presented for MAF<0.01 vs MAF>=0.01, low LD vs high LD results)? The contribution would be better quantified if authors made it clear what is the actual proportion of markers with such property. In addition, if low LD markers contribute more (proportionally?), would not it invalidate the LDSC approach (and support LDAK)? For robustness, I’d like to see whether the h² estimates are confirmed by a method that is based on the LDAK model. I do not want to impose too much extra work, but this is the most central point of the paper. If it is such a model trait, would be informative to know how these two methods compare and if there are differences why we see them?

Re: We used GREML-LDMS h^2_{SNP} to partition h^2_{SNP} based on MAF and LD). Our h^2_{SNP} analyses focused on GREML methods, not LDSC. The LDSC h^2_{SNP} estimates were provided in the manuscript as a QC step. This is because genetic correlation estimates may be unreliable when LDSC h^2_{SNP} analyses are too low.

We appreciate the referee’s comment on showing h^2_{SNP} on a finer grid. Unfortunately, we are unable to perform these analyses due to computational constraints; alternatively, this would be at the expense of power (due to a GRM based upon a smaller experimental sample size).

We have added details of the markers used for each GREML-LDMS component in **Supplementary Table 2**. We have added p-values for tests of enrichment in the main text (**Results, page 5**), with further details on the calculation within **Methods (page 21)**. The tests for enrichment found that there was significant enrichment for common variants (fold-enrichment = 1.44, $p=4.0e-71$) in low LD regions (fold-enrichment=1.40, $p=4.8e-42$).

As suggested, we also performed an LDAK analysis, which gave $h^2_{SNP} = 0.38$ (SE = 0.01). This compares well with our estimate of 0.39 (SE = 0.01). We also refer to [Evans et al. 2018 “Comparison of methods that use whole genome data to estimate the heritability and genetic architecture of complex traits” Nature Genetics], which found that GREML-LDMS is more robust than LDAK with respect to h^2_{SNP} estimation.

Have the authors performed LDSC stratified by functional categories?

Re: We thank the reviewer for this helpful suggestion. We have since performed the LDSC cell-type enrichment. The main text has been edited (**page 6**) and figures have been added (**Supplementary Figure 3b**). After multiple test correction, we found enrichment for conserved regions ($p=7.6e-8$), histone modifications (H3K27ac, H3K4me1, H3K4me3, H3K9ac; $1.2e-17 < p < 1.7e-7$), super enhancers ($p=3.5e-11$), DNase hypersensitivity sites (DHS; $p=2.8e-8$), digital genomic footprint (DGF; $p=5.7e-7$) and transcription factor binding sites (TFBS; $1.61e-6$) (**page 6, Supplementary Note 1**).

While overall pleiotropy (LDSC result) was quite weak, could the authors identify most prominent pleiotropy among the top hits (GW-significant)?

Re: We thank the reviewer for this comment. The aim of our PRS analysis was to identify pleiotropy among the top hits. However, we have since decided to drop this analysis for clarity.

We performed an additional analysis using ICSNPathway to identify genes and pathways that the COJO GWS SNPs may be associated with, which may provide clues to pleiotropy amongst the GWS SNPs (**page 7, Supplementary Note 1**). This analysis identified enrichment for 85 candidate causal pathways, including many developmental processes. We focused specifically on the pathways titled “Reproduction” and “Reproduction Pathways”, and performed GeneCards lookups on the genes grouped within these pathways. These provided gene-level support for the interesting genetic correlations that we identified using LDHub. There was evidence for TGF-beta pathway involvement, which links mesenchymal tissues, androgen pathways and bone mineral density; HPA axis involvement, which is relevant to adrenarche and early-onset puberty (**page 13**); calcium homeostasis, relating to bone mineral density, as well as factors involved in sex differentiation and fertility (**Supplementary Note 1**).

I feel that the bit on selection is quite thinly covered in the manuscript. “It appears that MPB autosomal genetic variance is mostly driven by common variants in regions of low linkage disequilibrium, consistent with weak negative selection on MPB-associated loci” I do not see any strong quantified evidence presented for this claim, but a list of qualitative statements (mostly common variants, low LD [except chrX], (insignificant but) negative genetic correlation with fathered offsprings). I feel this statement being too general and true for almost any complex trait. Could the authors either expand on this angle or tone it down.

Re: We thank the reviewer for this helpful comment. We have opted to tone this angle down, and the manuscript has been updated in the Abstract (**page 1**) and Discussion (**pages 15,16,17**). We have also performed tests to quantify the degree of h^2_{SNP} enrichment among common loci and low LD regions (**pages 5, 21**). Specifically, genetic variation for MPB was enriched among common variants ($h^2_{SNP}=0.405$, $SE=0.007$, $fold-enrichment=1.443$, $p=3.99e-71$) and in low LD regions ($h^2_{SNP}=0.29$, $SE=0.01$, $fold-enrichment=1.399$, $p=4.81e-42$).

Minor comments:

1. “We identified 54,107 autosomal GWS loci” - I believe these are not “loci”, but SNPs.

Re: We have updated the manuscript with this correction (**page 5**).

2. “The trait “number of children fathered” had a weak phenotypic correlation with age-adjusted MPB” – out of how many tested traits?

Re: We only calculated phenotypic correlation between number of children fathered and MPB. This is because the genetic correlation between these two traits was very low, and we were concerned that the genetic correlation results were unreliable with $h^2_{SNP} < 0.05$. Hence, we estimated the phenotypic correlation as a proxy, since often phenotypic and genetic correlations are of similar magnitude. We have clarified this on **page 8**.

3. “There is some evidence of interactions between the loci on the autosomes and the X-chromosome. Separate 241 conditional analyses on each of the autosomal and X-chromosome sets of SNPs gave 705 and 35 conditionally-independent SNPs respectively” – I do not see why this would point to interaction. Could not it simply be some correlation between chrX and autosomal markers? (E.g. through uncorrected pop stratification) Having 5 more independent loci when looking at them independently is not really solid evidence; may not even be significant? It is also contradicted by the fact that the bivariate GCTA showed no significant covariance between the chrX and autosomal GRMs when fitted for MBP.

Re: After submission, we realised that using the --actual-geno flag for the COJO analysis was giving potentially unreliable results as COJO is not designed to accept the X-chromosome. We found that using summary-statistics gave more robust results for the X-chromosome, the trade-off being some loss of power. Our COJO results have been updated accordingly (**pages 5-6, Figure 3**), and as a brief comparison, the differences are shown below.

COJO analysis	Aggregated	Autosomes	ChrX
--cojo-actual-geno flag	735 (711 aut + 24 chrX)	705	35
Summary statistics	626 (597 aut + 29 chrX)	597	26

In addition, we have removed the aggregated analysis results from the manuscript, leaving only the results from the analyses for autosomes and the X-chromosome alone. This is because GCTA-COJO is not programmed to accept the X-chromosome in an aggregated analysis. More specifically, the 0/2 coding of X-chromosome genotypes becomes problematic when performing calculations in the aggregated analysis, but these issues cancel out when the X-chromosome is considered alone.

This has had minor flow-on implications throughout the manuscript, namely the annotation of the X-chromosome loci (**Supplementary Note 1, Supplementary Table 5**), and the variance estimate using the X-chromosome COJO SNPs (**page 11, Figure 2, Supplementary Table 1**).

4. The BOLT-LMM analysis was carried out in 205K samples (out of which >150K are certainly unrelated), but the GCTA was done on 75K unrelated samples only. Why?

Re: We used different sample sizes in BOLT-LMM versus GCTA-GREML analyses due to computational constraints. Our justification and methods for this are provided in the **Methods** section, under “Narrow-sense heritability (h^2) estimation”, **paragraph 3 (page 19)**. Essentially, we aimed to maximise the number of individuals with relatives (estimated relatedness from SNP data > 0.05) so as to estimate both variance components with good precision. This meant including 47,957 individuals with relatives in the data, and a random sample of 40,000 singletons (individuals with no relatives).

Reviewer #2 (Remarks to the Author):

Yap et al. Estimation and Dissection of Genetic Variation for Male Pattern Baldness

The authors have used UKB to do a GWAS of MPB, and argue that it is a model trait. They find the pedigree-heritability to be 0.62 and SNP-heritability to be 0.39. They find 735 loci and these tend to have weak negative selection. The authors devote some focus to the X-chromosome, as this is often ignored in GWAS of MPB and other traits. Here it is somewhat simplified due to the fact that MPB is a sex-limited trait.

The authors are well-versed in heritability/GWAS methods (having invented some of them) and there are few issues with the manuscript.

Re: We thank the reviewer for their comments, and they have improved the readability of our manuscript. We wish to draw attention to the fact that we have amended our SNP selection analyses (using GCTA-COJO), finding a total of 626 SNPs instead.

After submission, we realised that using the `--actual-geno` flag for the COJO analysis was giving potentially unreliable results as COJO is not designed to accept the X-chromosome. We found that using summary-statistics gave more robust results for the X-chromosome, the trade-off being some loss of power. Our COJO results have been updated accordingly (**pages 5-6, Figure 3**), and as a brief comparison, the differences are shown below.

COJO analysis	Aggregated	Autosomes	ChrX
<code>--cojo-actual-geno</code> flag	735 (711 aut + 24 chrX)	705	35
Summary statistics	626 (597 aut + 29 chrX)	597	26

In addition, we have removed the aggregated analysis results from the manuscript, leaving only the results from the analyses for autosomes and the X-chromosome alone. This is because GCTA-COJO is not programmed to accept the X-chromosome in an aggregated analysis. More specifically, the 0/2 coding of X-chromosome genotypes becomes problematic when performing calculations in the aggregated analysis, but these issues cancel out when the X-chromosome is considered alone.

This has had minor flow-on implications throughout the manuscript, namely the annotation of the X-chromosome loci (**Supplementary Note 1, Supplementary Table 5**), and the variance estimate using the X-chromosome COJO SNPs (**page 11, Figure 2, Supplementary Table 1**).

Major comments:

1) It would be very helpful in the results section to define how you've modeled MPB as a trait. The reader has to dig through the paper to identify that you have generated adjusted MPB scores after residualizing for age, centre, ethnicity and PC's.

Re: We thank the reviewer for bringing this to our attention, and the manuscript has been edited to improve readability. We have included a section "Modelling of the MPB trait" within the Results section (**page 4**). We took as our trait residuals from regressing the MPB scores on age, assessment centre, ethnicity and 40 principal components, calculated using the UKB European population.

2) While the authors have used 4 ordered categories to ascertain the concept of MPB, MPB is clearly not 4 simple ordered categories (rather it is a continuously progressive trait that varies by rate of progression across men). Early onset MPB affects a smaller number of men, whereas nearly all men experience some hair loss later in life. This likely reflects different biology. I therefore have a difficult time agreeing with the authors that MPB is a "model trait" in the same way that height is a model trait. Further arguments against this concept include the high degree of variance explained by the X chromosome, due to the large effect size common variants present there. Few traits that I'm aware of have such a disproportionate effect of the X chromosome.

Re: We thank the reviewer for pointing out this issue. We have edited the manuscript to reduce the emphasis on using MPB as a "model trait" in the Abstract (**page 1**) and Discussion (**page 17**).

With regards to the reviewer's comment on early- versus late-onset baldness " ... likely reflect[ing] different biology", we believe that early-onset MPB may simply be a more severe manifestation of the condition. For example, [Liu et al. 2016 "Prediction of male-pattern baldness using genotypes" EJHG] found that early-onset MPB has greater SNP liability than late-onset MPB. This may imply that MPB severity is related to the burden of MPB-associated loci. Specifically:

- Early-onset MPB: autosomal common $h^2_{\text{SNP}}=0.558$ (SE=0.216) versus chrX common $h^2_{\text{SNP}}=0.233$ (SE=0.011)
- Late-onset MPB: autosomal common $h^2_{\text{SNP}}=0.424$ (SE=0.234) versus chrX common $h^2_{\text{SNP}}=0.098$ (SE=0.049).

In addition, we demonstrated that the frequency of the MPB-associated trait-increasing alleles increases approximately linearly with MPB severity [Yap et al. 2018 "Misestimation of heritability and prediction accuracy of male-pattern baldness" Nature Communications]. This is also consistent with MPB severity being related to the genetic burden, and that the trait is essentially quantitative (although data collection used a discrete, ordinal scale).

3) It would be interesting to understand if the genetic correlation analyses (LDSC and PRS) correlate with sexual activity. I'm not certain which measures of sexual activity are present in UKB, but this could easily be explored.

Re: We thank the reviewer for suggesting this analysis. We have added an additional analysis, inputting our summary statistics into LDHub (now containing over 800 GWAS, including hundreds from the UKB) to gain wholesale insights into genetic correlations for over 800 traits. None of the traits identified at a $p<0.05$ threshold were related to sexual activity. Reassuringly, all of the UK Biobank traits that were identified by the analysis with $h^2>0.05$ had already been tested in our study. The updated results are on **page 8**, and an extensive discussion on the genetic correlations that we identified is provided on **pages 12-15**. Succinctly, these analyses provided further evidence to support an association with earlier age of puberty onset ($r_g=-0.18$, $p=1.7e-11$ for facial hair onset; $r_g=-0.11$, $p=1.6e-5$ for voice breaking onset; $r_g=-0.09$, $p=2.2e-8$ for menarche; $r_g=-0.16$, $p=1.0e-4$ for difference in height from age eight to adult), increased bone mineral density ($r_g=0.15$, $p=3.2e-5$ for lumbar spine) and pancreatic β -cell function ($r_g=0.15$, $p=3.0e-5$). These traits seem plausible: puberty-onset traits may converge upon an androgenetic pathway (**page 13**); bone mineral density may relate to the Wnt pathway, which is modulated by androgens (**pages 13-14**), and pancreatic β -cell function also appears to fit with a picture of metabolic disease associated with early-onset MPB (**pages 14**). Further discussion on potential pathways and function is provided in **Supplementary Note 1**.

We also note that we have also removed the PRS analysis section to clarify the message of the manuscript, in response to another reviewer's comment.

Minor comments:

1) It would be helpful to show the definitions of the MPB categories early in the manuscript and in Figure 1.

Re: We have edited the **Results** section as suggested, and have altered the **Figure 1** panel to include the diagrams of baldness patterns and corresponding scores. We have included a heading for "Modelling of the MPB trait" (**page 4**).

2) Most readers will not recall differences in metrics of heritability. While the authors have written lovely articles on these topics, it might be helpful to have a small box reminding the reader of the definitions.

Re: We thank the reviewer for this helpful suggestion, and we have added **Box 1 (page 29)**.

3) When the authors state that they have identified 54,107 autosomal GWS loci, might it make more sense to state 54,107 GWS SNPs?

Re: We have made this correction accordingly (**page 5**).

4) I don't think I understand why the number of COJO loci differs when the autosomal and X chr SNPs are analyzed separately and together. Could the authors explain this?

Re: As discussed at the beginning of our response, we have altered our COJO analysis to 1) analyse summary statistics and 2) to drop the aggregated analysis due to X-chromosome coding issues. Hence, the disparity in the number of COJO loci is now resolved.

5) It would be helpful if Table 1 contained the number of cases/controls for case/control analyses (i.e. prostate cancer)

Re: We have edited **Table 1** accordingly.

Reviewer #3 (Remarks to the Author):

The authors state their aim to study male patterned balding (MPB) as a 'model genetic trait'. They do this by performing the largest GWAS to date for this trait in the large UK Biobank (UKB) study, and intriguingly they describe the GWAS for MPB as "a by-product of our analysis" (Line 200). I presume this explains why they perform only very limited downstream analysis of the GWAS findings. I find this limits some of the claims made here, whereas if the focus is indeed on broader lessons regarding inheritance then those aspects need further consideration and interpretation.

Re: We thank the reviewer for their thoughtful comments, and as a result we have extended our downstream analyses. We also expanded our genetic correlation analysis to LDHub, with interesting results (pages 8, 12-15, 24, Supplementary Table 9). We used FUMA analyses to generate list of potentially relevant genes and pathways (pages 6-7, 22, Supplementary Note 1, Supplementary Tables 13-14). We also performed LDSC partitioned h^2 analyses to look for enrichment by cellular elements, as well as by cell-type groups (page 6, Supplementary Note 1, Supplementary Table 12, Supplementary Table 14). We performed ICSNP analyses to identify genes associated with the top GWS COJO SNPs (page 6, Supplementary Note 1)

Comments:

1. A key limitation to MPB being a 'model genetic trait' is the lack of agreement in this field regarding trait measurement and analysis, which leads substantial inconsistencies in heritability estimates and GWAS signals. Although UKB ordered the categories 1-4, the difference between categories 2 (vertex) and 3 (crown) are qualitative rather than quantitative. The age distribution in Fig 1 appears to indicate similar distributions for groups 1-2, while groups 3-4 show a separate close distribution. Genetics could shed important light on this issue. Are groups 2 and 3 relatively distinct? Are 1-2 and 3-4 genetically similar groups?

Re: We recently wrote a correspondence [Yap et al. 2018 "Misestimation of heritability and prediction accuracy of male-pattern baldness" Nature Communications], which addresses the issue of groups 2 and 3 being treated as a continuum, and justifies our use of MPB as a continuous trait. Briefly, we took the 47 most associated independent autosomal loci identified outside the UKB dataset, and plotted the frequency of the trait-increasing allele within each of the MPB categories. As shown in the below graph, the allele frequency shows an approximately linear increase across the MPB scores.

2. The claimed relevance of MPB to reduced fertility (Abstract Line 15) requires much further substantiation. This claim appears to be based only on a weak ($P=0.01$) genetic correlation with number of live children in women and a weak (and surprisingly isolated) phenotypic correlation with “number of children fathered”. Genetic and phenotypic associations between MPB and these traits should be described more systematically in both men and women.

Re: We thank the reviewer for their suggestion. We have downplayed the emphasis on reduced fertility and negative selection in the Abstract (**page 1**), and have made some changes to the Discussion with respect to the discussion of negative selection to better integrate our findings (**page 16**). The updated LDHub results provided interesting findings relating to early-onset puberty that we now focus on, as well as associations with bone mineral density and metabolic traits.

3. The relevance of MPB to other traits is also very limited. Table 1 claims to show genetic correlations with female- and male-limited traits. Several of the (presumed, due to lack of clear labelling) 'female-limited traits' are also relevant to males (e.g. age at 1st/last birth, BW of child, number of children). Several other 'epidemiological links' are listed in Para 1 of the main text and should also be tested.

Re: We thank the reviewer for their comment. Although traits such as birth weight may be related to male factors, we believe that they can primarily be considered in the context of female fertility. For example, maternal predisposition and response to stress (psychological and physiological), cardiometabolic disease (eg. gestational diabetes), placental function and substance use are under genetic influence, and these are particularly critical in establishing a foetal environment that may cause variation in birth weight. The 'female-limited traits' are listed in **Methods** under the **Female data** paragraph (**page 19**).

With regards to the other “epidemiological links”, we uploaded our results to LD Hub to gain wholesale insights into genetic correlations, and the manuscript has been changed accordingly (**pages 8, 12-15, 24, Supplementary Table 9**). Reassuringly, all of the UK Biobank traits that were identified by the analysis with $h^2 > 0.05$ had already been tested in our study. The updated results are on **page 8**, and an extensive discussion on the genetic correlations that we identified is provided on **pages 12-15**. Succinctly, these analyses provided further evidence to support an association with earlier age of puberty onset ($r_g = -0.18$, $p = 1.7e-11$ for facial hair onset; $r_g = -0.11$, $p = 1.6e-5$ for voice breaking onset; $r_g = -0.09$, $p = 2.2e-8$ for menarche; $r_g = -0.16$, $p = 1.0e-4$ for difference in height from age eight to adult), increased bone mineral density ($r_g = 0.15$, $p = 3.2e-5$ for lumbar spine) and pancreatic β -cell function ($r_g = 0.15$, $p = 3.0e-5$). These traits seem plausible: puberty-onset traits may converge upon an androgenic pathway (**page 13**); bone mineral density may relate to the Wnt pathway, which is modulated by androgens (**pages 13-14**), and pancreatic β -cell function also appears to fit with a picture of metabolic disease associated with early-onset MPB (**pages 14**). Further discussion on potential pathways and function is provided in **Supplementary Note 1**.

We note that LDHub analyses did not find genetic correlations between MPB and previously-described epidemiological associations (for prostate cancer, and coronary artery disease), and the h^2_{SNP} estimate for type 2 diabetes was too low for the estimate to be reliable ($h^2_{SNP} = 0.036$, $SE = 0.003$) (**page 15**).

4. The genetic overlap between MPB and puberty timing appears novel. However, the proposed explanation and discussion (Line 244-252) need further consideration. They say that the link is "easily reconciled" but immediately then indicate the opposite by stating that the traits rely on different sources of androgens. Which tissues are enriched by expression of genes located near to MPB associated SNPs? Is there a causal relationship between these traits?

Re: We thank the reviewer for their comment. Although MPB and puberty onset may rely on different androgen sources (testicular vs adrenals, respectively), the underlying genetic variants that contribute to these traits may be similar as androgens are the mediating ligand. These variants plausibly explain the genetic correlation seen here. We have updated the manuscript to clarify this statement (**page 14**). We show results from a Mendelian Randomisation analysis using the GSMR tool to test for the putative causal effect of age of menarche (using summary statistics from 3740 GWS SNPs from the UKB GWAS that we performed; note that there is no sample overlap as both traits are sex-limited) on MPB. However, we have opted not to include these results in the paper due to length concerns, and also because the traits occur in different sexes so one cannot “cause” the other.

Exposure	Outcome	bxy	se	p	nsnp
Age of menarche	MPB	-0.0392	0.0061	9.4e-11	199

We expanded our downstream analysis to check for tissues enriched by gene expression near MPB-associated SNPs FUMA GWAS (which draws upon the GTEx dataset). The manuscript has been amended to address this (**pages 6-7, 22, Supplementary Note 1, Supplementary Figures 4-5**).

5. Regarding the key claim of weak negative selection on MPB-associated loci, Fig 2A shows more heritability explained by low LD vs. High LD components of the genome, but to assess whether there is enrichment requires information on the definition, size and numbers of variants in each component. Analyses should quantify the degree and probability of enrichment.

Re: We thank the reviewer for their suggestion. We have added details of the markers used for each LDMS component in **Supplementary Table 2**. We have added p-values for tests of enrichment in the main text (**page 5**), with further details on the calculation within **Methods (page 21)**, and have modified the discussion (**page 10**). Specifically, genetic variation for MPB was enriched among common variants ($h^2_{\text{SNP}}=0.405$, SE=0.007, fold-enrichment=1.443, $p=3.99e-71$) and in low LD regions ($h^2_{\text{SNP}}=0.29$, SE=0.01, fold-enrichment=1.399, $p=4.81e-42$).

We have also made some additions to the discussion of negative selection on **page 16**.

6. They identify 14 'independent signals' in the 2.3 MB Androgen receptor gene region at Xq12. Because the association signal here is so strong it is difficult to establish whether these are truly independent. Fig 3C should focus more precisely on the X chromosome to more clearly display this region and indicate LD with the strongest Xq12 SNP.

Re: We thank the reviewer for this suggestion. We have added a LD heatmap panel in **Figure 3d** to better demonstrate the LD structure within the Xp11.2 and Xq13.2 bands. This illustrated that most of the X-chromosome SNPs were indeed independent, with the exception of two pairs that were in high LD but remained jointly significant due to their large effect sizes. We also note that the updated COJO analysis meant that our X-chromosome results have changed slightly (**pages 5-6, 11, Supplementary Table 5**) with respect to the SNPs and loci that were identified.

7. Line 74-75 "These results may be inflated by common environment effects and non-additive genetic variation". This statement is vague and should be expanded here and commented on further in the discussion. Do they infer that specifically the brother-brother pairs h^2 estimate is inflated by CE? Inclusion of non-additive genetic variation does not 'inflate' heritability, rather its omission in father-son pairs is 'deflation'. Comment on which estimate is nearer the truth?

Re: We thank the reviewer for this comment. We have edited the manuscript to clarify that the brother-brother h^2_{SNP} estimate specifically may be inflated by common environment (**page 8**). We suggest that the brother-brother estimate may give the more relevant estimate given that the X-chromosome exerts a large effect, and that father-son pairs do not share the X-chromosome.

8. 'Interaction' between autosomal and X-linked loci is intriguing and needs further comment. What do you mean by interaction and how could this impact GCTA models? Is there correlation between autosomal and X-PAR variants for some reason?

After submission, we realised that using the `--actual-geno` flag for the COJO analysis was giving potentially unreliable results as COJO is not designed to accept the X-chromosome. We found that using summary-statistics gave more robust results for the X-chromosome, the trade-off being some loss of power. Our COJO results have been updated accordingly (**pages 5-6, Figure 3**), and as a brief comparison, the differences are shown below.

COJO analysis	Aggregated	Autosomes	ChrX
<code>--cojo-actual-geno</code> flag	735 (711 aut + 24 chrX)	705	35
Summary statistics	626 (597 aut + 29 chrX)	597	26

In addition, we have removed the aggregated analysis results from the manuscript, leaving only the results from the analyses for autosomes and the X-chromosome alone. This is because GCTA-COJO is not programmed to accept the X-chromosome in an aggregated analysis. More specifically, the 0/2 coding of X-chromosome genotypes becomes problematic when performing calculations in the aggregated analysis, but these issues cancel out when the X-chromosome is considered alone.

This has had minor flow-on implications throughout the manuscript, namely the annotation of the X-chromosome loci (**Supplementary Note 1, Supplementary Table 5**), and the variance estimate using the X-chromosome COJO SNPs (**page 11, Figure 2, Supplementary Table 1**).

Thus, we have removed the paragraph on “interactions” as it is no longer valid.

Reviewer #1 (Remarks to the Author):

The authors very carefully addressed (almost) all my comments. There is only one, where I feel that more effort could have been made to convince me:

“We appreciate the referee’s comment on showing h^2 SNP on a finer grid. Unfortunately, we are unable to perform these analyses due to computational constraints; alternatively, this would be at the expense of power (due to a GRM based upon a smaller experimental sample size).”

I’m not sure why it would be computationally so much more demanding to have a finer grid resolution for the MAF or LDscore? Surely, there would be loss of power in the individual h^2 estimation, but eventually some (e.g. weighted least squares) functions would be fitted to the estimates to test trends, which would be highly interesting.

Reviewer #2 (Remarks to the Author):

The authors have satisfied my comments.

Reviewers' comments:

Reviewer #1 (Remarks to the Author):

The authors very carefully addressed (almost) all my comments. There is only one, where I feel that more effort could have been made to convince me:

“We appreciate the referee’s comment on showing h^2 SNP on a finer grid. Unfortunately, we are unable to perform these analyses due to computational constraints; alternatively, this would be at the expense of power (due to a GRM based upon a smaller experimental sample size).”

I’m not sure why it would be computationally so much more demanding to have a finer grid resolution for the MAF or LDscore? Surely, there would be loss of power in the individual h^2 estimation, but eventually some (e.g. weighted least squares) functions would be fitted to the estimates to test trends, which would be highly interesting.

We thank the reviewer for their constructive comment. Our computing infrastructure and quotas have undergone recent changes, which has enabled us to perform these analyses using LDMS.

We used a total of 12 bins: four LD score quartiles, and 3 MAF bins ($1.5e-5 < \text{MAF} \leq 1e-3$; $1e-3 < \text{MAF} \leq 0.01$; $0.01 < \text{MAF} < 0.5$). We also included more rare variants (an extra bin with $1.5e-5 < \text{MAF} < 1e-3$), which added over 14.5 million extra SNPs. We also changed the LD score calculation to the individual-SNP genome-wide approach (versus our previous regional calculations per chromosome), as this has recently been shown to be superior [Evans et al. 2018 Comparison of methods that use whole genome data to estimate the heritability and genetic architecture of complex traits]. Supplementary Data 1 and 2 have been updated, and the results are copied below (Supplementary Data 2 also contains the number of SNPs used in each component):

	1st LD quartile		2nd LD quartile		3rd LD quartile		4th LD quartile		Row sum	s.e.
	Est	s.e.	Est	s.e.	Est	s.e.	Est	s.e.		
$1.5 \times 10^{-5} \leq \text{MAF} < 1 \times 10^{-3}$	-0.013054	0.023157	-0.011023	0.019781	0.003954	0.005903	-0.003854	0.001803	-0.023977	0.024464
$1 \times 10^{-3} < \text{MAF} < 0.01$	0.004909	0.011947	0.001332	0.009988	-0.000799	0.007424	-0.000937	0.002118	0.004505	0.015335
$0.01 \leq \text{MAF} < 0.5$	0.151313	0.007906	0.149125	0.005679	0.073691	0.004119	0.060237	0.002925	0.434366	0.008294
Column sum	0.143168	0.026677	0.139434	0.022378	0.076846	0.010137	0.055446	0.003933		
Total sum	0.414894	0.027969								
Log Likelihood	-42212.182									
Sample size	75654									

The gist of our findings remain the same: overwhelming enrichment among common variants ($h^2_{SNP,LDMS,common} = 0.43$, $SE = 0.008$, fold-enrichment = 3.63, $p = 3.6e-177$) and a trend towards lower LD regions ($h^2_{SNP,LDMS,lowLD} = 0.283$, $SE = 0.03$, fold-enrichment = 1.36, $p = 8.7e-3$). Common variants essentially explained all of the variance, with negligible contribution from rare variants. (Note that the no-constraint option was used in the analysis, which gave negative h^2 for the $1.5e-5 < \text{MAF} \leq 1e-3$ category, explaining why $h^2_{SNP,LDMS,common}$ exceeded $h^2_{SNP,LDMS}$.) There was also a declining trend across LD quartiles: quartiles 1 to 4 contributed 35%, 34%, 19% and 13% of h^2_{SNP} , respectively (Supplementary Data 2). There was relatively similar h^2_{SNP} between the 1st/2nd and 3rd/4th LD quartiles, but less so between the 2nd/3rd quartiles.

The corresponding parts of the manuscript has been updated throughout (changes highlighted in yellow), including Figure 2 and Supplementary Data 1 and 2. We have also made minor changes to align with the editorial policy checklist (e.g. making the manuscript more succinct while retaining the core message).

Reviewer #2 (Remarks to the Author):

The authors have satisfied my comments.

Reviewer #1 (Remarks to the Author):

I really appreciate that the authors went into great depth to address my comment. I am fully convinced now.